# pYtags enable spatiotemporal measurements of receptor tyrosine kinase signaling in living cells

Payam E Farahani[1], Xiaoyu Yang[2,3], Emily V Mesev[4], Kaylan A Fomby[4], Ellen H Brumbaugh-Reed[4,5], Caleb J Bashor[2,6]*, Celeste M Nelson[1,4]*, Jared E Toettcher[4]*

[1]Department of Chemical & Biological Engineering, Princeton University, Princeton, United States; [2]Department of Bioengineering, Rice University, Houston, United States; [3]Program in Systems, Synthetic, and Physical Biology, Rice University, Houston, United States; [4]Department of Molecular Biology, Princeton University, Princeton, United States; [5]IRCC International Research Collaboration Center, National Institutes of Natural Sciences, Tokyo, Japan; [6]Department of Biosciences, Rice University, Houston, United States

*For correspondence:
caleb.bashor@rice.edu (CJB);
celesten@princeton.edu (CMN);
toettcher@princeton.edu (JET)

Competing interest: The authors declare that no competing interests exist.

**Abstract** Receptor tyrosine kinases (RTKs) are major signaling hubs in metazoans, playing crucial roles in cell proliferation, migration, and differentiation. However, few tools are available to measure the activity of a specific RTK in individual living cells. Here, we present pYtags, a modular approach for monitoring the activity of a user-defined RTK by live-cell microscopy. pYtags consist of an RTK modified with a tyrosine activation motif that, when phosphorylated, recruits a fluorescently labeled tandem SH2 domain with high specificity. We show that pYtags enable the monitoring of a specific RTK on seconds-to-minutes time scales and across subcellular and multicellular length scales. Using a pYtag biosensor for epidermal growth factor receptor (EGFR), we quantitatively characterize how signaling dynamics vary with the identity and dose of activating ligand. We show that orthogonal pYtags can be used to monitor the dynamics of EGFR and ErbB2 activity in the same cell, revealing distinct phases of activation for each RTK. The specificity and modularity of pYtags open the door to robust biosensors of multiple tyrosine kinases and may enable engineering of synthetic receptors with orthogonal response programs.

## Editor's evaluation

Your study provides strong and convincing evidence that pYtags enable spatiotemporal measurements of receptor tyrosine kinase signaling in living cells. This is highly significant as it can be used to study in real-time receptor signaling in healthy and diseased cells.

## Introduction

Development and homeostasis of multicellular organisms require that cells sense and respond to diverse microenvironmental signals. Receptor tyrosine kinases (RTKs) are one widely expressed class of cell-surface receptors that play a key role in this information processing (*Lemmon and Schlessinger, 2010*). RTK activation is triggered by growth factors or hormones that bind to the extracellular domains of receptors, inducing conformational changes that lead to receptor dimerization and the autophosphorylation of tyrosine residues in the C-terminal tails. Phosphorylated RTKs, hereafter described as 'activated' RTKs, present phosphotyrosine-containing motifs that bind to downstream effectors that

signal to multiple pathways. RTKs are thus positioned as the uppermost node of a complex network of intracellular signaling.

RTK signaling provides a high-dimensional input space through which this class of receptors regulates a variety of cellular processes (*Lemmon and Schlessinger, 2010*). Human cells express at least 58 RTKs whose binding partners vary widely, so the pathways that are activated in a particular cell depend on the set of receptors expressed on its surface (*Madhani, 2001*; *Salokas et al., 2022*). Some classes of receptors can form homo- or heterodimers, further diversifying signaling responses (*Kanakaraj et al., 1991*; *Harari and Yarden, 2000*; *Del Piccolo et al., 2017*). Moreover, a single RTK can bind to many different ligands that are capable of inducing distinct conformational changes to tune downstream signaling dynamics (*Freed et al., 2017*; *Hu et al., 2022*). These observations underscore the substantial complexity present at even the uppermost node of the RTK signaling network: the interaction between receptors and their ligands.

Recent advances in live-cell biosensors have enabled the study of intracellular signaling at unprecedented spatiotemporal resolution. Signaling responses can be measured with a temporal precision on the order of seconds, and recent studies have described approaches to multiplex biosensors of multiple pathways in a single cell (*Regot et al., 2014*) or to use barcoding strategies to monitor many biosensors within a population of cells (*Kaufman et al., 2022*; *Yang et al., 2021*). Yet the development of biosensors for specific RTKs has been somewhat limited. A fluorescent Grb2-based biosensor has been widely used to measure general RTK activity but does not distinguish between the many receptors that can recruit Grb2 (*Reynolds et al., 2003*). FRET-based biosensors of epidermal growth factor receptor (EGFR) (*Komatsu et al., 2011*; *Sorkin et al., 2000*; *Itoh et al., 2005*; *Kurokawa et al., 2001*) and platelet-derived growth factor receptor (PDGFR) (*Seong et al., 2017*) have been described, but rely on components of endogenous substrates that participate in interactions with multiple RTKs, and their specificity remains to be characterized. A more specific Src homology 2 (SH2)-based biosensor for EGFR was recently reported but required extensive engineering and may compete with other SH2 domain-containing proteins for binding to phosphotyrosine motifs required for signaling (*Tiruthani et al., 2019*). Consequently, the field still lacks modular live-cell biosensors to monitor the activity of any specific RTK of interest.

Here, we describe pYtags, a versatile and modular RTK biosensing strategy. In the pYtag approach, the C-terminus of a user-defined RTK is labeled with a tyrosine activation motif that, when phosphorylated, binds selectively to a fluorescently labeled tandem SH2 (tSH2) reporter. This binding results in the depletion of the fluorescent reporter from the cytosol and local accumulation at membranes where the receptor is activated. We show that an EGFR pYtag biosensor quantitatively reports on the activity of EGFR with undetectable crosstalk from other unlabeled RTKs. We use pYtags to quantify EGFR activation in response to various ligands, revealing ligand- and dose-specific signaling dynamics. Mathematical modeling and experimental validation suggest that different ligands affect the signaling dynamics of EGFR by altering the dimerization affinity of ligand-bound receptors. We further demonstrate that pYtags can be applied to other receptors, including FGFR1, PDGFRβ, VEGFR3, and the ligand-less RTK ErbB2, a challenging case study due to its need to signal through receptor heterodimers. We develop a second, orthogonal pYtag that we use to simultaneously monitor EGFR and ErbB2 activity within the same cell. Finally, we demonstrate that pYtags can be inserted into the genome using CRISPR/Cas9-based editing, which enables reporting of endogenous receptors and eliminates confounding effects from ectopic expression. pYtags thus offer a modular strategy for measuring the activity of specific RTKs in individual living cells and may serve as a platform for engineering phosphotyrosine-based cargo that can be recruited to synthetic receptors.

## Results

### An orthogonal tag for monitoring tyrosine phosphorylation events

Many biosensors of kinase activity are based on two components: a synthetic substrate that is predominantly phosphorylated by a single kinase of interest, and a domain that binds specifically to the phosphorylated substrate. To apply similar logic in the design of an RTK biosensor, it would be desirable to introduce a tyrosine-containing peptide that can be uniquely phosphorylated by an RTK of interest and a binding domain that interacts only with the phosphorylated peptide. Fulfilling these requirements is especially challenging in the case of tyrosine kinases, which typically lack one-to-one

specificities between tyrosine kinase, tyrosine substrate, and SH2-binding domains (*Miller, 2003*). However, immune cells offer examples of highly specific phosphotyrosine signaling. During activation of the T-cell receptor (TCR), pairs of tyrosine residues located in immunoreceptor tyrosine-based activation motifs (ITAMs) are phosphorylated and serve as sites for the selective recruitment of the tandem SH2 domain of ZAP70 (ZtSH2), which binds 100-fold more tightly than individual ZAP70 SH2 domains with the same ITAMs and displays minimal crosstalk with other phosphotyrosine-containing peptides (*Figure 1A*; *Isakov et al., 1995*; *Labadia et al., 1996*; *Wange et al., 1993*). Because neither the TCR nor ZAP70 are expressed by nonimmune cells, we reasoned that an ITAM/ZtSH2 pair may be repurposed as an orthogonal interaction module to monitor the activity of a desired RTK in nonimmune contexts. If ITAMs appended to the C-terminal tail of an RTK are phosphorylated upon activation of the receptor, then this should create a selective binding site for a fluorescently labeled ZtSH2 (*Figure 1B*). At the cellular level, such a ZtSH2-based reporter should localize to the cytosol when RTK signaling is low and be recruited to the cell membrane when RTK signaling is high (*Figure 1B*).

We first applied this biosensing approach to detect the activation of EGFR, the most well-characterized of the 58 known human RTKs (*Salokas et al., 2022*). In an initial screening experiment, we tested six ITAMs from the CD3γ, CD3δ, CD3ε, and CD3 ζ chains of the TCR. In each case, we fused three identical repeats of the ITAM to the C-terminal tail of EGFR followed by the FusionRed fluorescent protein (EGFR-ITAM-FusionRed). We co-expressed each receptor with an iRFP-labeled ZtSH2 (iRFP-ZtSH2) in NIH3T3 fibroblasts (*Figure 1B*), which express very low levels of endogenous EGFR (*Di Fiore et al., 1987*; *Livneh et al., 1986*; *Eierhoff et al., 2010*). In cells expressing EGFR-ITAM-FusionRed, treatment with epidermal growth factor (EGF) resulted in rapid translocation of ZtSH2 from the cytosol to the cell membrane (*Figure 1C*, *Figure 1—video 1*), which we assessed by quantifying the percentage of ZtSH2 cleared from the cytosol (see 'Methods'). Clearance of ZtSH2 from the cytosol was quickly reversed by treatment with gefitinib, an inhibitor of EGFR kinase activity, indicating that ZtSH2 can serve as a rapid, reversible reporter of EGFR activation (*Figure 1C*). Cells lacking EGFR or expressing an ITAM-less EGFR-FusionRed exhibited no change in ZtSH2 localization in response to treatment with EGF or gefitinib (*Figure 1D and E*), demonstrating that the ZtSH2 reporter responds specifically to the activation of an ITAM-labeled RTK. Although all six ITAMs that we tested appear capable of functioning as biosensors of RTK activity, we chose to focus on the CD3ε ITAM for all subsequent experiments due to its reported selectivity for ZtSH2 over other phosphotyrosine-binding domains (*Osman et al., 1995*; *Ravichandran et al., 1993*; *Love and Hayes, 2010*). We refer to the resulting two-component biosensor as a phosphotyrosine tag (pYtag): a tyrosine activation motif that recruits its complementary tSH2 reporter to report on signaling.

Grb2-based biosensors have been used extensively to monitor receptor tyrosine kinase activity (*Reynolds et al., 2003*). However, Grb2 binds to many activated RTKs, making this biosensing strategy unable to specifically detect activity of a single RTK of interest. To compare our ITAM-labeling approach to Grb2-based biosensing, we simultaneously expressed a fluorescent Grb2 construct (Grb2-TagBFP) and iRFP-ZtSH2 in NIH3T3 cells expressing either EGFR-ITAM-FusionRed or an ITAM-less EGFR-FusionRed construct (*Figure 1—figure supplement 1A*). Stimulating these cells with EGF revealed that Grb2 localized to the membrane in both ITAM-tagged and ITAM-less EGFR cells, whereas ZtSH2 only showed membrane recruitment in the ITAM-tagged context (*Figure 1—figure supplement 1B–E*). These data further confirm that ZtSH2 is specific for ITAM-tagged RTKs, a feature that is not shared by Grb2-based reporters.

Since pYtags introduce additional tyrosine residues and SH2-containing peptides to an RTK signaling complex, we asked whether this biosensing strategy interferes with signaling downstream of EGFR. We stimulated NIH3T3 cells expressing either EGFR-FusionRed or EGFR pYtag (EGFR-CD3ε-FusionRed; iRFP-ZtSH2) with EGF and measured signaling responses as a function of time by immunoblotting (*Figure 1F*). We found that EGFR, Erk, and Akt were phosphorylated at similar levels in the two cell lines (*Figure 1G*), suggesting that ITAM phosphorylation and ZtSH2 recruitment do not interfere with signaling downstream of EGFR.

In T cells, CD3 ITAMs are typically phosphorylated by Src family kinases (SFKs) upon engagement of TCRs (*Gaud et al., 2018*). Because EGFR also activates SFKs, we reasoned that pYtags could be phosphorylated indirectly by EGFR-associated SFKs rather than by the kinase domain of EGFR itself (*Figure 1H*). To test whether SFK-mediated phosphorylation is required for pYtag function, we expressed EGFR pYtag in mouse embryonic fibroblasts lacking all three ubiquitously expressed SFKs:

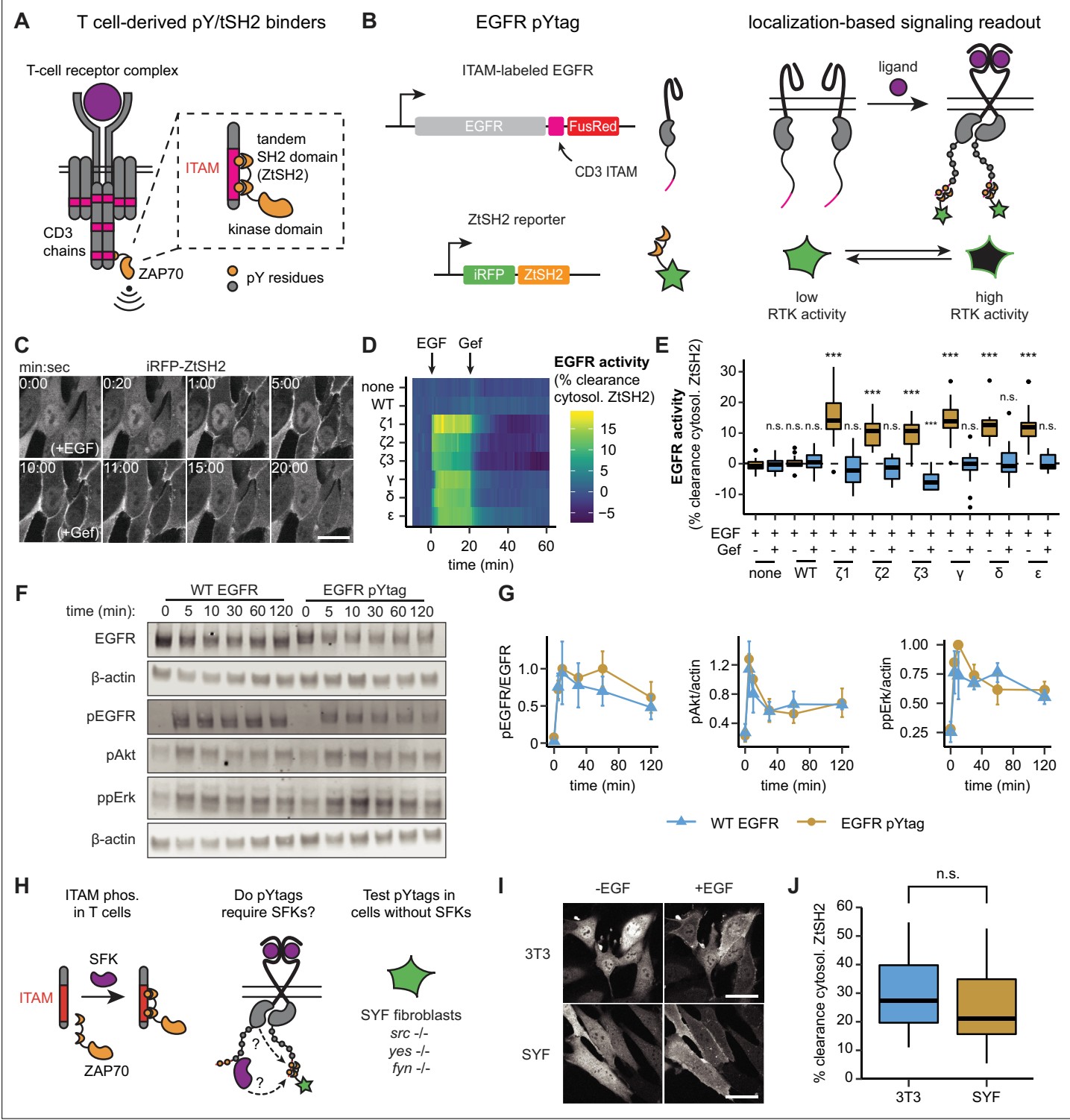

**Figure 1.** pYtags: a biosensing strategy to monitor receptor tyrosine kinase (RTK) activity in living cells. (**A**) The T-cell receptor complex contains six immunoreceptor tyrosine-based activation motifs (ITAMs) from CD3 chains that, when phosphorylated, bind to the tSH2 domain of ZAP70 (ZtSH2). (**B**) Three repeats of CD3 ITAMs were appended to the C-terminus of epidermal growth factor receptor (EGFR) and clearance of ZtSH2 from the cytosol was assessed. (**C**) Timelapse images of NIH3T3 cells expressing EGFR pYtag (CD3ε variant), treated first with EGF (100 ng/mL) and then with gefitinib (10 μM). Scale bar, 20 μm. (**D**) Mean clearance of cytosolic ZtSH2 in cells co-expressing iRFP-ZtSH2 and EGFR C-terminally labeled with one of six CD3 ITAMs. EGF (100 ng/mL) and gefitinib (10 μM) were sequentially added at times denoted by arrows. n = 2 independent experiments. (**E**) Clearance of cytosolic ZtSH2 10 min post-EGF treatment and 40 min post-gefitinib treatment from (**D**). Lines denote mean values, boxes denote 25–75th percentiles,

*Figure 1 continued on next page*

*Figure 1 continued*

and whiskers denote minima and maxima. n ≥ 14 cells from two independent experiments. n.s., not significant, ***p<0.001 by Kolmogorov–Smirnov test with cells expressing no additional EGFR 10 min post-EGF. (**F**) Immunoblots of NIH3T3 cells expressing either WT EGFR or EGFR pYtag treated with EGF (100 ng/mL). (**G**) Mean ± SEM levels of EGFR, Akt, and Erk phosphorylation from (**F**). n = 3 independent experiments. (**H**) The EGFR pYtag was tested in SYF cells to determine whether SFKs are required for ITAM phosphorylation. (**I**) Representative images of NIH3T3 and SYF cells expressing EGFR pYtag, treated with EGF (100 ng/mL). Scale bars, 40 µm. (**J**) Mean clearance of cytosolic ZtSH2 in SYF and NIH3T3 cells 10 min after treatment with EGF. For each condition, n > 20 cells from three independent experiments. See also *Figure 1—video 1*.

The online version of this article includes the following video, source data, and figure supplement(s) for figure 1:

**Source data 1.** Uncropped gels for *Figure 1F*.

**Figure supplement 1.** Grb2 fails to discriminate between immunoreceptor tyrosine-based activation motif (ITAM)-labeled and unlabeled RTKs.

**Figure supplement 2.** Effects of the expression levels of pYtag components in NIH3T3 cells.

**Figure supplement 3.** Effects of the expression levels of pYtag components in HEK293T cells.

**Figure supplement 4.** Mathematical modeling of the effects of pYtag component expression levels on biosensor readout.

**Figure 1—video 1.** Timelapse of iRFP-ZtSH2 in NIH3T3 cells co-expressing iRFP-ZtSH2 and EGFR-CD3ε-FusionRed.
https://elifesciences.org/articles/82863/figures#fig1video1

Src, Yes, and Fyn (SYF cells) (*Figure 1H*; *Klinghoffer et al., 1999*). As in NIH3T3 cells, SYF cells exhibited strong clearance of ZtSH2 from the cytosol after stimulation with EGF (*Figure 1I and J*). Combined with the rapid responses observed after EGFR stimulation and inhibition (*Figure 1C–E*), these results suggest that the EGFR pYtag acts as a direct biosensor of EGFR activity.

## Assessing reporter activity as a function of the expression levels of pYtag components

Our pYtag biosensor relies on two components: an ITAM-tagged RTK and a fluorescent, cytosolic ZtSH2 probe. We reasoned that the measured output – the extent of ZtSH2 redistribution from cytoplasm to membrane – could depend on the expression level of both components, reminiscent of the responses observed for cellular optogenetics tools that re-localize between cytosol and membrane (*Toettcher et al., 2011*). To further investigate this dependency, we measured ZtSH2 membrane translocation while simultaneously quantifying the initial levels of membrane-associated EGFR and cytoplasmic ZtSH2 in individual cells by live imaging. These experiments were carried out in two cell lines used throughout our study: NIH3T3 cells expressing EGFR-CD3ε-FusionRed and iRFP-ZtSH2 (*Figure 1—figure supplement 2*) and HEK293T cells expressing EGFR-CD3ε-mNeonGreen and mScarlet-ZtSH2 (*Figure 1—figure supplement 3*).

When comparing cells with similar levels of ZtSH2 but different levels of EGFR, we observed a greater degree of ZtSH2 redistribution in cells with higher EGFR expression (*Figure 1—figure supplements 2A and 3A*). Conversely, imaging cells with similar levels of EGFR but different levels of ZtSH2 revealed more complete ZtSH2 clearance in cells with lower ZtSH2 expression (*Figure 1—figure supplements 2B and 3B*). More broadly, quantifying cytosolic ZtSH2 clearance across all cells revealed consistently higher degrees of ZtSH2 clearance in cells with a high EGFR:ZtSH2 expression ratio, as measured by the ratio of their fluorescence intensities prior to stimulation (*Figure 1—figure supplements 2C,D and 3C,D*). This phenomenon was also matched by a simplified mathematical model of EGFR activation and ZtSH2 redistribution (see text for Figure 3 and 'Methods' for a detailed discussion of the model), where the percentage translocation of ZtSH2 varied linearly with the EGFR-to-ZtSH2 ratio until it saturated under conditions where the amount of ZtSH2 exceeded that of active EGFR (*Figure 1—figure supplement 4*). In sum, these results suggest that a simple ratio – the amount of ITAM-tagged EGFR relative to the amount of cytoplasmic ZtSH2 – sets the dynamic range of this translocation-based biosensor.

We also tested whether the EGFR:ZtSH2 ratio might also affect the kinetics of translocation. We first quantified responses across all cells and sorted the resulting heatmaps in order of increasing EGFR:ZtSH2 ratio (*Figure 1—figure supplements 2E and 3E*), which again confirmed that ZtSH2 clearance depended on this ratio. We next min–max normalized each response, enabling us to compare response kinetics to be compared in an amplitude-independent manner (*Figure 1—figure supplements 2F and 3F*). While steady-state ZtSH2 cytosolic clearance depended on the EGFR:ZtSH2 ratio, we observed similarly rapid activation kinetics at all expression levels.

These results suggest that the pYtag biosensor approach faithfully reports on the kinetics of RTK activation across a range of component expression levels, but that response amplitude should be compared only between cells at similar expression levels of both the ITAM-tagged RTK and fluorescent ZtSH2 probe. Our results also suggest guiding principles for future cell line engineering. Ideally, a translocation-based biosensor would exhibit measurable redistribution between cytosol and membrane, yet avoid completely clearing from the cytosol at low stimulus levels. Our results provide an intuitive way to achieve this response – matching the expression of the ZtSH2 cytosolic component to be proportional to the expression of the ITAM-tagged receptor. For example, if one desires to detect activity of RTKs that are expressed at low levels, it is important to engineer cells with correspondingly low ZtSH2 expression to observe potent redistribution of this biosensor.

## pYtags reveal the dynamics of EGFR signaling at high spatiotemporal resolution

Because ZtSH2 can in principle be recruited rapidly on and off the membrane or other subcellular compartments where active receptors are found, we reasoned that pYtags could be used to monitor the subcellular activity of an RTK in individual cells over time. To test this possibility, we first performed rapid, high-resolution imaging of NIH3T3 cells treated with EGF (*Figure 2A*). We found that ZtSH2 was cleared from the cytosol in a biphasic manner, with an initial phase of rapid cytosolic clearance occurring within the first ~40 s, followed by a further gradual increase in cytosolic clearance over the subsequent ~20 min (*Figure 2B*). We observed sustained ZtSH2 translocation to the membrane for at least 30 min, indicating that high levels of membrane-localized, active EGFR are persistent at the membrane under these conditions (*Figure 2A*). Similar biphasic responses for EGFR were observed previously at the population-level using a split luciferase system (*Macdonald-Obermann and Pike, 2014*; *Macdonald-Obermann et al., 2012*) and in vitro for EGFR on synthetic vesicles (*Kovacs et al., 2015*), but have never been reported for individual cells.

EGFR is well known to be heavily regulated by intracellular trafficking through processes such as internalization, degradation, and recycling back to the cell membrane (*Sorkin and Goh, 2009*). However, we observed continued ZtSH2 and EGFR membrane localization in NIH3T3 pYtag-expressing cells for at least 30 min after EGF stimulation (*Figure 2A*), indicating that endocytosis does not result in the rapid clearance of EGFR from the cell membrane in these cells, possibly due to saturation of receptor endocytosis at high expression levels (*Lund et al., 1990*). To gain more insight into trafficking in our system, we first validated that endocytosis of EGFR is not altered by expression of the novel pYtag components. We stimulated NIH3T3 cells expressing either EGFR-FusionRed or the pYtag system (EGFR-CD3ε-FR; iRFP-ZtSH2) and stained for the early endocytic marker EEA1 at various time points after stimulation with 100 ng/mL EGF (*Figure 2—figure supplement 1A*). Quantifying EGFR membrane intensity revealed that cells retain high levels of membrane-associated EGFR for at least 60 min after EGF stimulation, and that this response is independent of whether our pYtag components are present (*Figure 2—figure supplement 1B and C*).

Despite sustained, high levels of membrane-localized EGFR, we also observed that pYtag-expressing cells stimulated with EGF contained puncta that were positive for both total EGFR and ZtSH2 (*Figure 2C and D*). Co-staining cells at various time points after EGF stimulation for the early endocytic marker EEA1 revealed colocalization between puncta containing EGFR, ZtSH2, and EEA1 in both NIH3T3 and HEK293T cells (*Figure 2—figure supplement 1D–G*). Furthermore, subsequent treatment with gefitinib rapidly eliminated ZtSH2 from EGFR-positive puncta, suggesting that the enrichment of ZtSH2 at puncta is indicative of signaling from endosomal compartments (*Figure 2C and D*). These results are consistent with prior reports that internalized EGFR remains bound to its ligand and can transmit signals from endosomal compartments (*Haugh et al., 1999*). Nevertheless, we find that EGFR-overexpressing NIH3T3 cells retain a high degree of membrane-associated receptor signaling for at least 1 hr after EGF stimulation.

At the tissue scale, the spatiotemporal distribution of RTK signaling can also be regulated by several processes including paracrine signaling (*De Simone et al., 2021*), morphogen gradients (*Casanova and Struhl, 1989*; *Sprenger and Nüsslein-Volhard, 1992*), and the mechanical properties of the local microenvironment (*Farahani et al., 2021*). We therefore asked whether pYtags could be used to monitor RTK signaling in multicellular contexts. We previously found that MCF10A human mammary epithelial cells form multicellular clusters when cultured on soft substrata and exhibit a

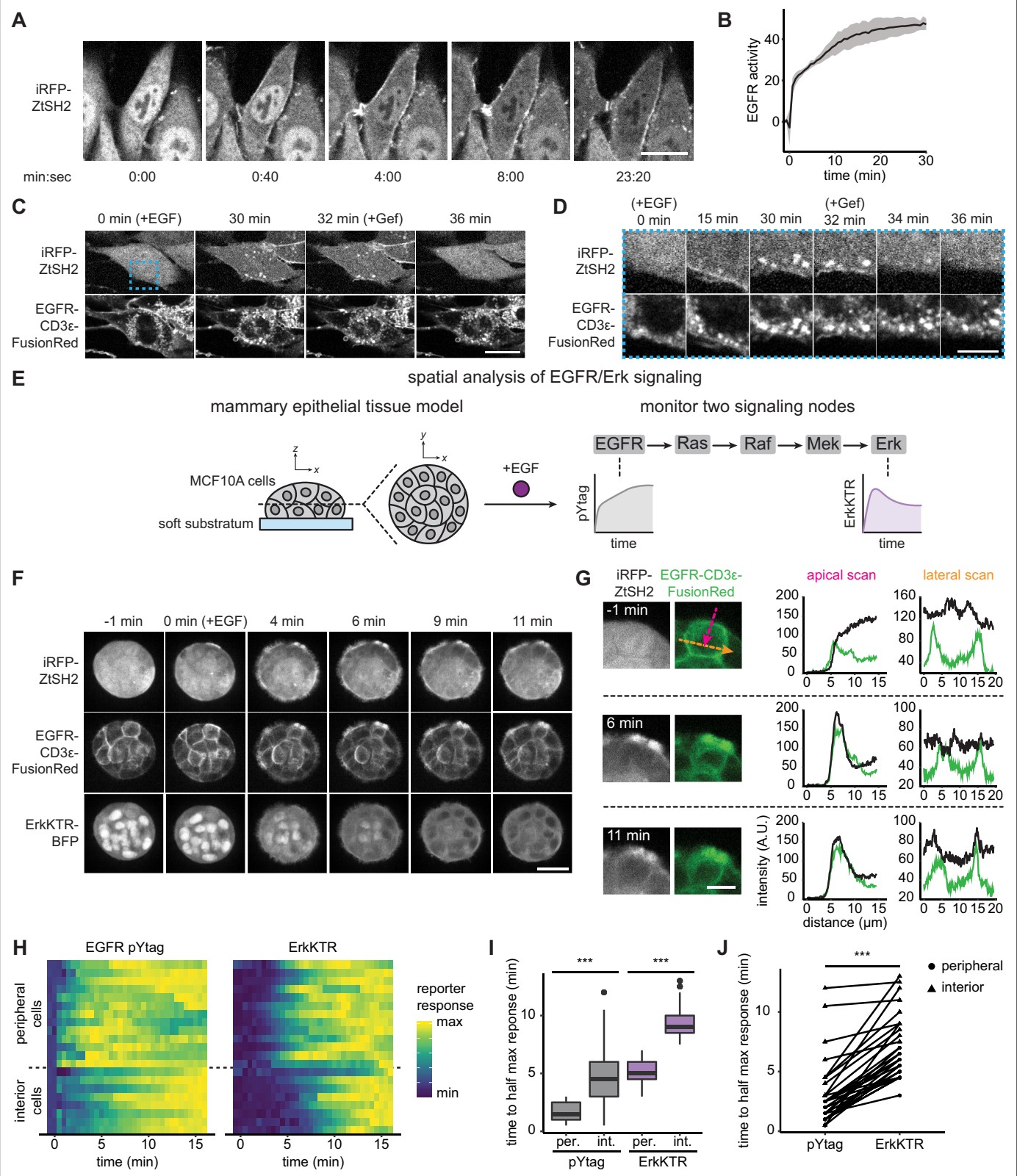

**Figure 2.** Monitoring epidermal growth factor receptor (EGFR) signaling at subcellular and multicellular length scales. (**A**) Images of EGFR pYtag-expressing NIH3T3 cells treated with EGF (20 ng/mL). Scale bar, 20 μm. (**B**) Mean ± SD clearance of cytosolic ZtSH2 from (**A**). n = 3 independent experiments. (**C**) EGFR pYtag-expressing NIH3T3 cells treated with EGF (20 ng/mL) were monitored for internalized ZtSH2-positive vesicles, and then treated with gefitinib (10 μM). Scale bar, 20 μm. (**D**) Timelapse images from the region denoted by the blue dashed border from (**C**). Scale bar, 10 μm.

*Figure 2 continued on next page*

*Figure 2 continued*

(**E**) MCF10A human mammary epithelial cells cultured on soft substrata form round, multilayered clusters. EGFR pYtag and ErkKTR were used to spatiotemporally monitor both EGFR and Erk responses after stimulation with EGF. (**F**) Images of MCF10A cells cultured on soft substrata and treated with EGF (100 ng/mL). Scale bar, 25 µm. (**G**) Apical and lateral enrichment of ZtSH2 was quantified by line scans denoted by magenta and orange vectors, respectively. Scale bar, 10 µm. (**H**) Heatmaps of EGFR pYtag and ErkKTR responses from (**F**). Rows denote individual cells. For each cell, signaling responses of each biosensor were normalized to their respective minima and maxima. (**I**) Time to half maximal response for EGFR pYtag and ErkKTR for cells from (**H**). n = 19 (periphery) and n = 11 (interior) cells from three biological replicates. (**J**) Time to half maximal response for EGFR pYtag and ErkKTR. Responses of individual cells are denoted by points and connected by lines. n > 30 cells from three biological replicates. ***p<0.001 by Kolmogorov–Smirnov test. See also *Figure 2—video 1*.

The online version of this article includes the following video and figure supplement(s) for figure 2:

**Figure supplement 1.** Analysis of epidermal growth factor receptor (EGFR) and ZtSH2 internalization.

**Figure 2—video 1.** Maximum intensity projection timelapse images of MCF10A cells co-expressing iRFP-ZtSH2 (left panel), EGFR-CD3ε-FusionRed (middle panel), and ErkKTR-BFP (right panel), cultured on soft substrata and treated with EGF (100 ng/mL).

https://elifesciences.org/articles/82863/figures#fig2video1

complex spatial pattern of EGF binding at cell membranes (*Farahani et al., 2021*). Measurements of fixed tissues revealed that EGF binds rapidly to the media-exposed membranes on the surface of the cluster but is excluded from lateral membranes and from cells located within the interior of the cluster (*Farahani et al., 2021*). To investigate the dynamics of this spatial pattern, we monitored EGFR signaling using two live-cell biosensors: EGFR pYtag and a kinase translocation reporter for downstream signaling through Erk (ErkKTR) (*Regot et al., 2014*). We treated MCF10A cells cultured on soft substrata with EGF and observed a radially directed wave of EGFR and Erk signaling: activation first appeared in cells at the periphery of the clusters before appearing in cells at the interior (*Figure 2F*, *Figure 2—video 1*). In cells at the periphery of clusters, we found that ZtSH2 was first enriched at the media-exposed membrane before localizing to lateral membranes (*Figure 2G*), highlighting the differences in receptor-level signaling between membrane subcompartments. Quantifying EGFR and Erk responses confirmed our qualitative observations, revealing a 2–4 min delay in EGFR and Erk signaling between the periphery and interior of clusters (*Figure 2H and I*). Notably, Erk responses also trailed those of EGFR by ~4 min, consistent with the delay in signal transmission previously observed from Ras to Erk (*Figure 2J*; *Toettcher et al., 2013*). These data support a model in which ligand–receptor interactions are limited at cell–cell contacts, producing an inward-traveling wave of RTK activation and downstream signaling. More broadly, our data demonstrate that pYtags can be used to reveal RTK signaling dynamics at high temporal resolution and over both subcellular and multicellular length scales.

## pYtags distinguish ligand- and dose-dependent signaling dynamics

We next set out to apply pYtags to an unresolved question in RTK signaling. While it has long been known that different RTKs regulate cellular processes through the dynamics of downstream signaling (*Freed et al., 2017*; *Marshall, 1995*; *Johnson and Toettcher, 2019*; *Santos et al., 2007*), recent evidence suggests that signaling dynamics can also differ between ligands for the same receptor (*Freed et al., 2017*). For instance, high-affinity EGFR ligands such as EGF produce long-lived EGFR dimers that are internalized and degraded, whereas low-affinity ligands such as epiregulin (EREG) and epigen (EPGN) produce shorter-lived EGFR dimers that prolong signaling, leading to distinct cell-fate outcomes (*Freed et al., 2017*; *Roepstorff et al., 2009*). Yet how these different ligands might alter minutes-timescale EGFR signaling dynamics in single cells remains unknown. We reasoned that pYtags would be ideal for measuring whether different EGFR ligands elicit distinct fast-timescale signaling dynamics.

We treated EGFR pYtag-expressing NIH3T3 cells with varying doses of either EGF or one of two low-affinity ligands, EREG or EPGN (see 'Methods'). For all ligands, we found that ligand stimulation was able to activate our pYtag system, driving ZtSH2 clearance that persisted for at least 30 min (*Figure 3—figure supplement 1*). The spatial distributions of ZtSH2 and EGFR were similar across all ligands, with both components enriched at the cell membrane throughout the stimulation timecourse (*Figure 3—figure supplement 1*). However, quantification of single-cell responses revealed that the kinetics of EGFR activation varied substantially between doses and ligands (*Figure 3A*). For EGF, we observed a gradual rise in activity at lower doses (0.2–2 ng/mL). This response switched to a biphasic

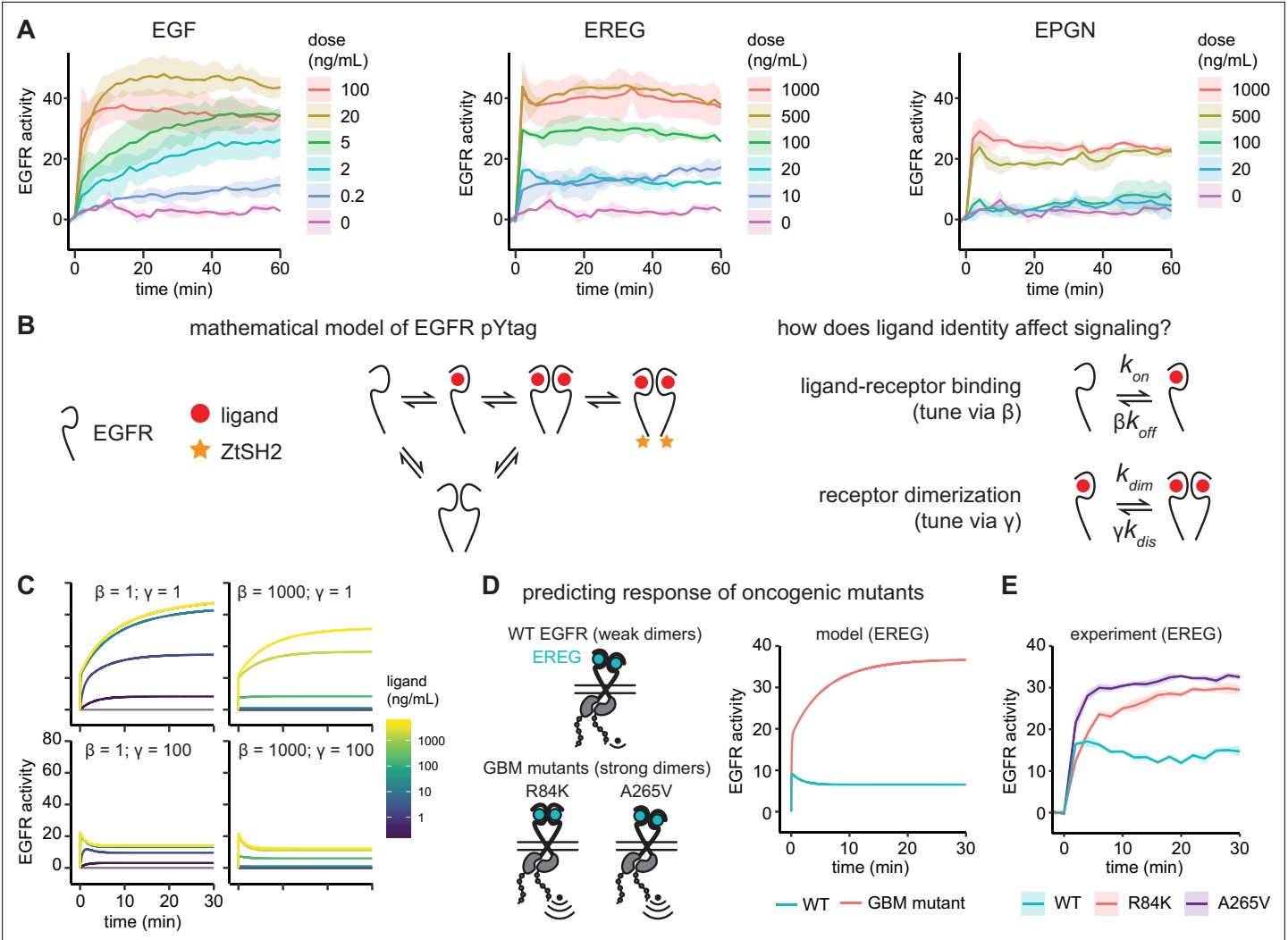

**Figure 3.** Epidermal growth factor receptor (EGFR) pYtag reveals dose- and ligand-dependent signaling dynamics. (**A**) Mean ± SD responses of EGFR pYtag-expressing NIH3T3 cells to varying doses of EGF, epiregulin (EREG), and epigen (EPGN). The same 0 ng/mL control was used for each ligand. Data were collected from 475 cells across four independent experiments with each dose tested at least twice. (**B**) Dose–response profiles from (**A**) were analyzed using a mathematical model of EGFR pYtag. (**C**) Simulations of EGFR pYtag responses to ligand of varying doses for different values of β and γ. (**D**) GBM-associated mutants of EGFR that form strong EREG-bound dimers were predicted to exhibit stronger pYtag responses to 20 ng/mL EREG compared to WT EGFR. (**E**) Mean ± SD pYtag response of WT and GBM-associated mutant EGFRs in NIH3T3 cells after EREG treatment (20 ng/mL). n = 3 independent experiments.

The online version of this article includes the following figure supplement(s) for figure 3:

**Figure supplement 1.** ZtSH2 and epidermal growth factor receptor (EGFR) localization in response to different ligands.

**Figure supplement 2.** Ligand-free dimers in mathematical model recapitulate biphasic signaling response of epidermal growth factor receptor (EGFR).

response at higher doses (5–100 ng/mL), with a rapid initial phase in first 1–2 min followed by a slower, continued rise (*Figure 3A*, left panel). In contrast, treatment with the low-affinity ligands EREG and EPGN produced a rapid rise in EGFR activity at all ligand doses, followed by a plateau (*Figure 3A*, middle and right panels). We also observed a small but reproducible transient peak of EGFR activation shortly after treatment with EREG and EPGN but not with EGF, indicative of a weakly adaptive response after stimulation with low-affinity ligands (*Figure 3A*, middle and right panels). These results demonstrate that different ligands produce markedly different signaling responses even in the first few minutes following stimulation. They also reveal an unusual property of the low-affinity RTK ligands EREG and EPGN: altering their concentrations has a strong effect on the amplitude, but not the kinetics, of EGFR activation.

What biochemical processes might lead to the dramatic differences in receptor activity kinetics observed between ligands? To begin addressing this question, we constructed a simplified mathematical model of the EGFR pYtag system based on mass-action kinetics that could be used to interrogate differences between ligands (*Figure 3B*). In our model, EGFR monomers can bind ligands, dimerize, undergo autophosphorylation and, lastly, bind to ZtSH2. We also modeled pre-formed EGFR dimers in the absence of ligand, based on several observations of these inactive dimers (*Bessman et al., 2014*; *Moriki et al., 2001*; *Yu et al., 2002*; *Saffarian et al., 2007*). In our model, these ligand-free dimers may also bind to ligands and become active. We chose to neglect endocytosis and trafficking in our model based on observations that our EGFR-overexpressing NIH3T3 cells maintain a large, constant pool of membrane-localized EGFR in each of our experiments (*Figure 1—figure supplement 2*, *Figure 2—figure supplement 1*, *Figure 3—figure supplement 1*). However, we expect that internalization and trafficking can play a crucial role in EGFR dynamics in many contexts, which would need to be included in future models to adequately assess those scenarios (*Sorkin and Goh, 2009*). Overall, our base model for EGF-induced signaling contains 16 equations (*Appendix 1—table 1*) and 9 parameters (*Appendix 1—table 2*). The values of six of the parameters were taken from previous work (*Macdonald and Pike, 2008*; *Schoeberl et al., 2002*; *Ottinger et al., 1998*), while the other three parameters reflect a lumped process of EGFR dimerization and activation and were set to match the EGF response kinetics observed in our experiments (see 'Methods'; *Figure 3—figure supplement 2*).

We next used this model to test how ligand-specific parameters might explain the differences in kinetics that we observed between EGF, EREG, and EPGN. Experimentally measured differences between ligands would be expected to alter two parameters in our base model representing EGF-binding (*Figure 3B*). First, high- and low-affinity ligands bind to EGFR with different affinities (*Freed et al., 2017*), this fold-change difference in binding affinity was set using a scaling parameter β (*Figure 3B*). Second, low-affinity ligands produce structurally different EGFR dimers compared to high-affinity ligands, thereby reducing the dimerization affinity of ligand-bound receptors (*Freed et al., 2017*; *Hu et al., 2022*). We modeled this fold-change in receptor dimerization affinity using a scaling parameter γ (*Figure 3B*). Prior experimental estimates suggest that high- and low-affinity ligands differ in both parameters over a 10- to 100-fold range (*Freed et al., 2017*; *Hu et al., 2022*); in our typical low-affinity ligand simulations, we set $\beta = 50$ and $\gamma = 100$ based on these prior reports.

Setting both β and γ to 1 (our base case, representing EGF) reproduced basic features of the EGF response, including a rapid initial phase of activation at high doses, followed by a gradual increase in activity at all doses (*Figure 3C*, top-left panel). Decreasing the dimerization affinity of ligand-bound receptors by increasing γ was sufficient to reproduce many of the features that varied with ligand identity. Simulations with $\gamma = 100$ produced an EREG/EPGN-like initial peak of receptor activation followed by a plateau (*Figure 3C*, lower panels). In contrast, simulating even a very large change in ligand binding affinity ($\beta = 1000$) was unable to qualitatively alter response kinetics (*Figure 3C*, right panel).

Some intuition can be gained by considering the differences in the dimerization affinity between inactive EGFR and ligand-bound EGFR in each simulated scenario. For our base model, EGF-bound receptors dimerize with higher affinity than ligand-less receptors, leading to a rapid initial phase of ligand-binding to ligand-less dimers, followed by a slower phase of dimerization and further activation of EGF-bound receptors. In the case of low-affinity ligands, a decreased affinity between ligand-bound receptors leads to a rapid steady state through activation of ligand-less dimers without gradual formation of additional complexes. In sum, our simulations suggest that differences in receptor dimerization affinity between EGFR ligands could play an important role in regulating receptor activation kinetics.

To test this prediction of our model, we experimentally altered the dimerization affinity of EGFR and monitored signaling responses using pYtags. We turned to glioblastoma multiform (GBM)-associated mutations in the extracellular domain of EGFR (R84K and A265V point mutations), which were recently shown to drive an ~650-fold increase in the dimerization affinity of EREG- and EPGN-bound receptors with only a 6-fold change to ligand-receptor binding (*Hu et al., 2022*). We first simulated these GBM-associated mutations in response to 20 ng/mL EREG, which led to the prediction of a stronger, more gradual signaling response in GBM-associated mutants compared to the wild-type receptor, reminiscent of the difference between EGF and EREG responses (*Figure 3D*). We then generated NIH3T3 cell lines expressing pYtagged EGFR variants harboring either the R84K or A265V mutation. Treating these cells with 20 ng/mL EREG produced signaling responses that closely matched the predictions of

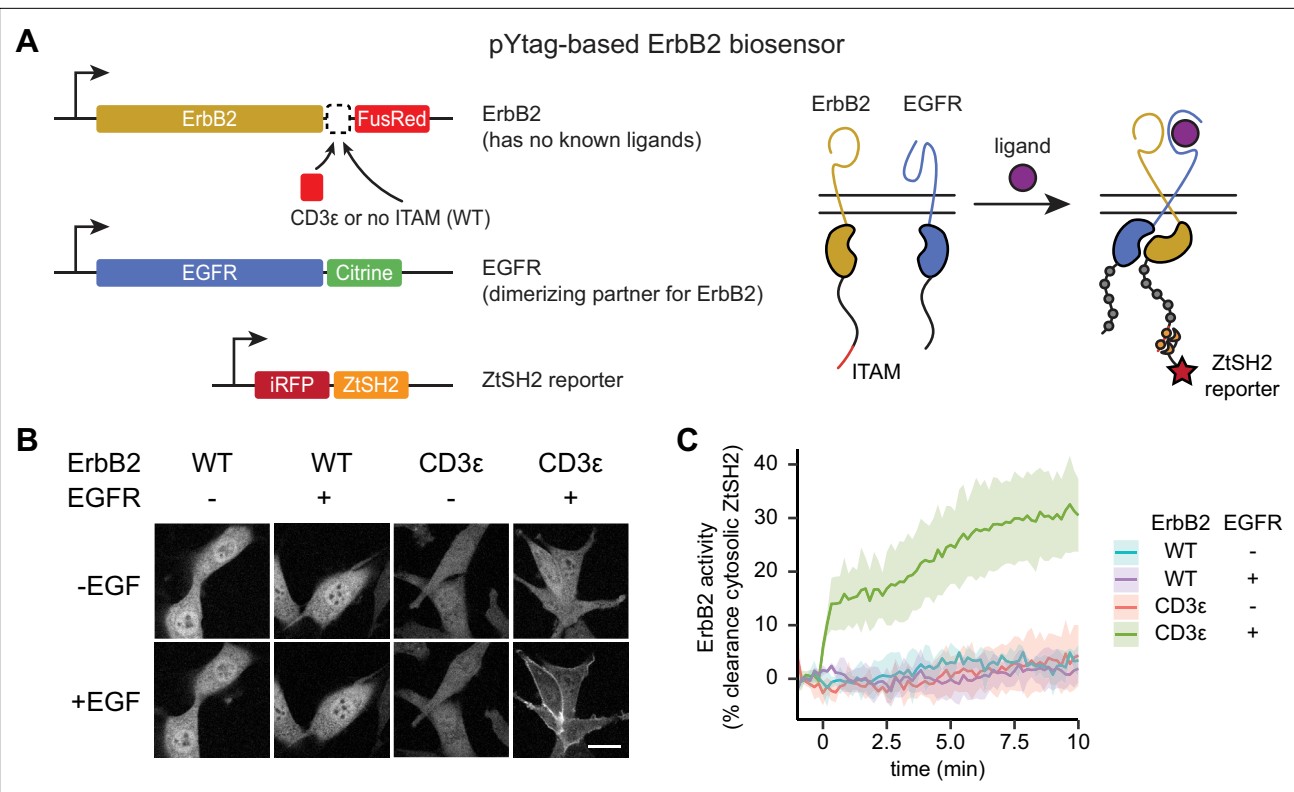

**Figure 4.** Monitoring distinct receptor tyrosine kinases (RTKs) in heterodimeric complexes. (**A**) In order to signal, the ligandless ErbB2 must heterodimerize with a ligand-binding member of the ErbB family. The pYtag strategy enables measurements of ErbB2's activity despite the co-activation of epidermal growth factor receptor (EGFR). (**B**) Representative images of NIH3T3 cells treated with EGF (100 ng/mL). Scale bar, 20 μm. (**C**) Mean ± SD clearance of cytosolic ZtSH2 after treatment with EGF (100 ng/mL). n = 3 independent experiments.

The online version of this article includes the following figure supplement(s) for figure 4:

**Figure supplement 1.** pYtag biosensors of additional receptor tyrosine kinases (RTKs).

our model: cells expressing WT EGFR exhibited a transient peak of activation and rapid plateau, while cells expressing GBM-associated mutants exhibited a stronger response that gradually increased over time (*Figure 3D and E*). Taken together, these data support the utility of the pYtag biosensor system for quantitatively studying the dynamics of receptor activation. Furthermore, our results suggest that dimerization affinity of ligand-bound EGFR is a key parameter that governs receptor activation.

## pYtags report the activity of distinct RTKs in heterodimeric complexes

One advantage of the pYtag approach is its modularity: ITAMs can in principle be fused to the C-termini of many different RTKs and recruit ZtSH2 upon stimulation. We thus tested whether pYtags could be adapted to monitor the activity of receptors other than EGFR. We designed pYtag biosensors for several additional RTKs: ErbB2, FGFR1, PDGFRβ, and VEGFR3. ErbB2 is a ligandless receptor and is particularly challenging to study because it can only be activated in conjunction with an additional ErbB family member such as EGFR (*Lemmon and Schlessinger, 2010*). As a result, general RTK biosensors such as those based on Grb2 (*Figure 1—figure supplement 1*) are unable to report specifically on ErbB2 activation dynamics due to the obligate presence of a second RTK (e.g., EGFR) in the same cells.

Starting from NIH3T3 cells expressing iRFP-ZtSH2, we generated cell lines expressing ITAM-labeled versions of each receptor (e.g., FGFR1-CD3ε-FusionRed), or a corresponding ITAM-less variant (e.g., FGFR1-FusionRed). For the ErbB2 case, we further transduced cells with EGFR-Citrine to express both required components of a functional receptor heterodimer (*Figure 4A*). In each case, we monitored ZtSH2 cytosolic depletion in cells after treatment with a canonical ligand for that pathway (e.g., FGF4,

PDGF-BB, or VEGF-C for FGFR1, PDGFRβ, and VEGFR3, respectively) in receptor variants with or without ITAMs.

We found that pYtags generalized well across all RTKs tested. Stimulating ErbB2 pYtag-expressing cells with EGF led to clearance of ZtSH2 from the cytosol, a response that required both the presence of ITAMs on ErbB2 and co-expression of EGFR (*Figure 4B and C*). Similarly, stimulating FGFR1, PDGFRβ, and VEGFR3 pYtag-expressing cells with their cognate ligands induced cytosolic clearance of ZtSH2, which did not take place in cells that expressed the ITAM-less variants (*Figure 4—figure supplement 1*). We observed FGFR1 and PDGFRβ clustering in internal compartments for both ITAM-tagged and untagged receptor variants (*Figure 4—figure supplement 1A and D*), possibly because of mislocalization due to receptor overexpression. However, even these cells retained ligand-dependent ZtSH2 redistribution from cytoplasm to membrane in an ITAM-dependent manner. Taken together, our data demonstrate that the pYtag approach is indeed modular and can be readily adapted for monitoring the activation of distinct RTKs in individual cells.

## Multiplexing RTK biosensors using orthogonal pYtags

Our pYtag design strategy relies on the high selectivity that can be produced by multivalent association between pairs of phosphotyrosine motifs and tSH2 domains. We thus hypothesized that two orthogonal pYtags could be deployed to monitor the activation of distinct RTKs in the same cell. To this end, we took advantage of another phosphotyrosine/tSH2 interaction pair from immune-specific signaling proteins to build a pYtag that functions orthogonally to the above-described CD3ε/ZtSH2 system. Following its binding to phosphorylated ITAMs on the TCR, ZAP70 phosphorylates non-ITAM tyrosine residues on the scaffold protein SLP76, which subsequently recruit multiple signaling components via SH2-mediated interactions (*Chakraborty and Weiss, 2014*; *Weiss and Littman, 1994*). Because immune signaling requires these events to be discrete, to be sequential, and to operate in proximity to each other, ZtSH2- and SLP76-recruited SH2s must have orthogonal binding specificities. We exploited this feature to engineer a second pYtag. We identified two tyrosine residues of SLP76, Y128 and Y145, which when phosphorylated bind to SH2 domains from the guanine nucleotide exchange factor Vav and the kinase ITK, respectively (*Figure 5A*; *Raab et al., 1997*; *Koretzky et al., 2006*; *Su et al., 1999*). To create a synthetic tSH2 that could bind tightly to the Y128/Y145 motif, we fused the SH2 domains of Vav and ITK together with a 10 bp glycine-serine linker to create VISH2. We then characterized the suitability of the SLP76/VISH2 system for monitoring RTK activity and quantified its crosstalk with the CD3ε/ZtSH2 system.

To determine whether the SLP76/VISH2 system functions as a pYtag biosensor in an orthogonal manner to CD3ε/ZtSH2, we generated NIH3T3 cells that stably co-expressed Clover-VISH2 and iRFP-ZtSH2; we then further transduced these cells with a variant of EGFR labeled with either three repeats of the E123-E153 region of SLP76 (EGFR-SLP76-TagBFP) or three repeats of the CD3ε ITAM (EGFR-CD3ε-TagBFP) (*Figure 5A*). Stimulating the two cell lines with EGF and monitoring the localization of VISH2 and ZtSH2 revealed that cells expressing SLP76-labeled EGFR exhibited clearance of VISH2 from the cytosol but no crosstalk with ZtSH2 (*Figure 5B–D*, *Figure 5—video 1*). Conversely, cells expressing CD3ε-labeled EGFR exhibited strong clearance of ZtSH2 from the cytosol with minimal crosstalk with VISH2 (*Figure 5B–D*, *Figure 5—video 1*). The VISH2/SLP76 and CD3ε/ZtSH2 pYtags thus operate independently of one another to report the activity of EGFR.

Having characterized an orthogonal pair of pYtags, we hypothesized that these biosensors could be used to simultaneously monitor the activation of both EGFR and ErbB2 in the same cell. To test this hypothesis, we generated NIH3T3 cells that co-expressed pYtags for EGFR (Clover-VISH2; EGFR-SLP76-TagBFP) and ErbB2 (iRFP-ZtSH2; ErbB2-CD3ε-FusionRed) (*Figure 5E*). We treated cells with 100 ng/mL EGF and observed that both VISH2 and ZtSH2 reporters cleared from the cytosol, as would be expected from the activation of EGFR and ErbB2 heterodimers. However, the biosensors revealed markedly different dynamics of receptor activation (*Figure 5F–H*, *Figure 5—video 2*). VISH2 cleared from the cytosol almost immediately (<30 s), indicating rapid phosphorylation of the EGFR C-terminal tail. In contrast, ZtSH2 exhibited a more gradual clearance from the cytosol (~3 min), suggesting a delay in ErbB2 phosphorylation. To verify that these dynamics are not due to the identity of the pYtag used to monitor each receptor or an artifact of the cell line expressing multiplexed biosensors, we compared the dynamics of EGFR and ErbB2 activation across experiments where NIH3T3 cells were treated with 100 ng/mL EGF, regardless of pYtag used (*Figure 5—figure supplement 1*).

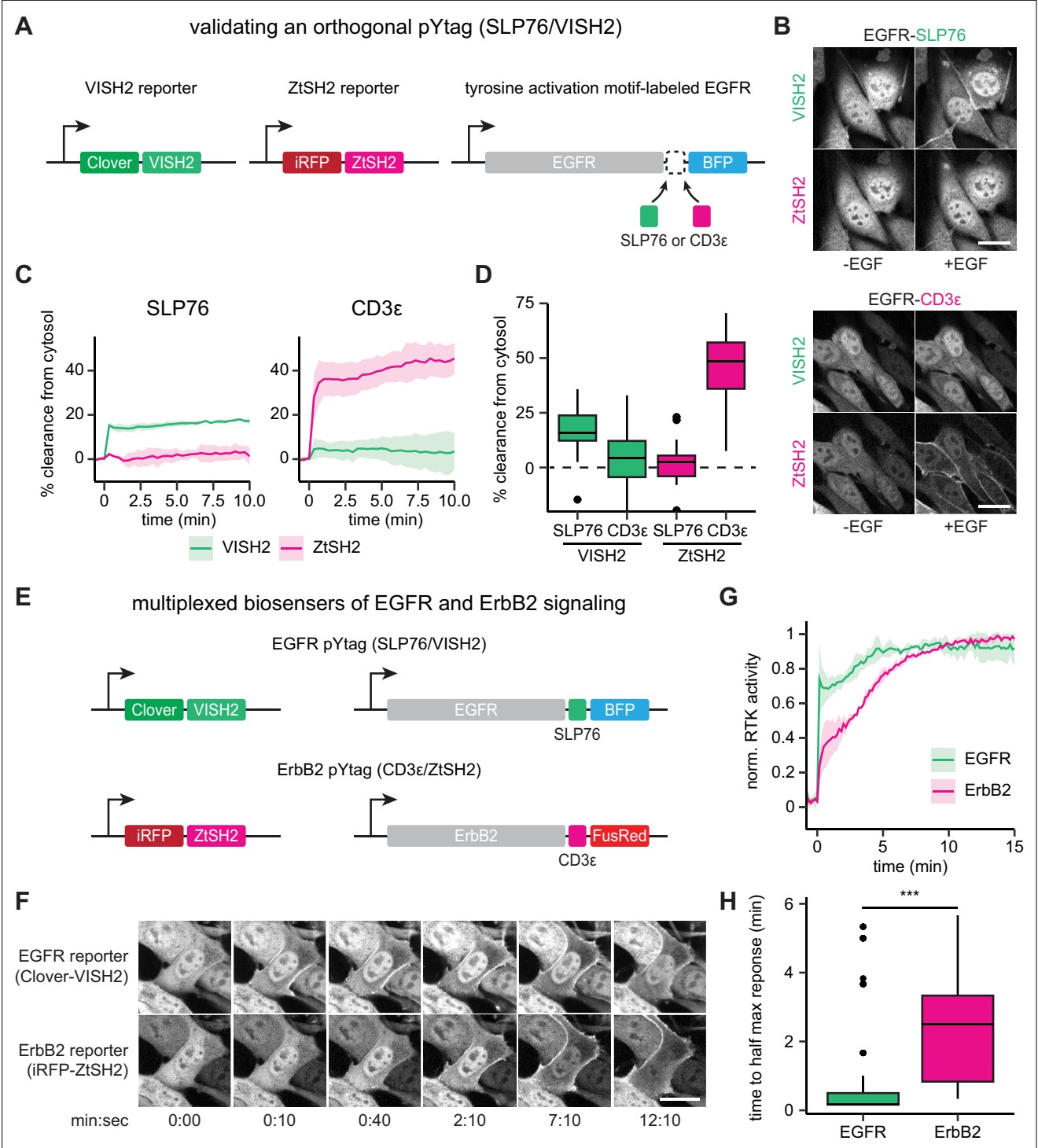

**Figure 5.** Orthogonal pYtags enable multiplexed receptor tyrosine kinase (RTK) biosensing. (**A**) To assess the performance of the VISH2/SLP76 system as a pYtag-based biosensor, VISH2 and ZtSH2 reporters were co-expressed in NIH3T3 cells along with either SLP76- or CD3ε-labeled epidermal growth factor receptor (EGFR). (**B**) NIH3T3 cells co-expressing VISH2 and ZtSH2 reporters before and 3 min after treatment with EGF (100 ng/mL). Scale bars, 20 μm. (**C**) Mean ± SD clearances of VISH2 and ZtSH2 from the cytosol, expressed with either SLP76- or CD3ε-labeled EGFR and stimulated with EGF (100 ng/mL). n = 3 independent experiments. (**D**) Response of VISH2 and ZtSH2 reporters 10 min after EGF treatment in (**C**). n > 30 cells from three independent experiments. (**E**) Orthogonal pYtags can be multiplexed to monitor the activity of multiple RTKs in the same cell. (**F**) Images of cells expressing orthogonal reporters for EGFR and ErbB2, treated with EGF (100 ng/mL). Scale bar, 20 μm. (**G**) Mean ± SD trajectories for EGFR and ErbB2 activity using multiplexed pYtags. For each reporter, the mean response was normalized to its minimum and maximum measured values. n = 3 independent experiments. (**H**) Time to half maximal response for individual cells from (**G**). Lines denote mean values, boxes denote 25–75th percentiles,

*Figure 5 continued on next page*

*Figure 5 continued*

and whiskers denote minima and maxima. n > 30 cells from three independent experiments. ***p<0.001 by Kolmogorov–Smirnov test. See also *Figure 5—video 1* and *Figure 5—video 2*.

The online version of this article includes the following video and figure supplement(s) for figure 5:

**Figure supplement 1.** Comparison of epidermal growth factor receptor (EGFR) and ErbB2 responses across experiments.

**Figure 5—video 1.** Timelapse of NIH3T3 cells co-expressing VISH2 and ZtSH2 reporters, and either SLP76- or CD3ε-labeled epidermal growth factor receptor (EGFR), treated with EGF (100 ng/mL).

https://elifesciences.org/articles/82863/figures#fig5video1

**Figure 5—video 2.** Timelapse of NIH3T3 cells co-expressing pYtag biosensors for epidermal growth factor receptor (EGFR) and ErbB2, treated with EGF (100 ng/mL).

https://elifesciences.org/articles/82863/figures#fig5video2

These included cases where EGFR was labeled with either VISH2 or ZtSH2, and cases where EGFR was expressed alone or in combination with ErbB2. We found that the absolute magnitude of tandem SH2 biosensor clearance varied substantially between cell lines (*Figure 5—figure supplement 1A*), consistent with our conclusion that the amplitude of the response varies with the EGFR:tSH2 expression ratio, which can be different between independently derived cell lines (*Figure 1—figure supplements 2–4*). However, when normalized to their maximum response amplitude, the dynamics of EGFR and ErbB2 activation were tightly overlapping in all cases, irrespective of the cell line or pYtag used (*Figure 5—figure supplement 1B and C*). These data provide further evidence that pYtag measurements faithfully reflect the activity dynamics of the RTK on which they report. Moreover, our results suggest that EGFR and ErbB2 are activated in distinct phases: EGFR is activated first, within seconds of stimulation, and then both EGFR and ErbB2 are further activated over the subsequent minutes.

## pYtags can be used to monitor the activity of endogenous RTKs

One drawback to the pYtag approach is that it requires the expression of a modified RTK on which ITAM sequences have been appended. We have shown throughout our study that ITAM-modified receptor variants can be ectopically expressed, but RTK overexpression is also well known to alter cell signaling dynamics, downstream pathway engagement, and cell fate outcomes (*Dikic et al., 1994*; *Traverse et al., 1994*). Conversely, the low expression levels of endogenous RTKs may only drive weak recruitment of tSH2, making biosensing by live-cell microscopy difficult. We thus asked whether pYtags could be adapted to monitor the activity of endogenously expressed RTKs.

We used CRISPR/Cas9-based genome editing to label the C-terminus of endogenous EGFR with three repeats of the CD3ε ITAM and a fluorescent protein via homology-directed repair (HDR) in HEK293T cells (*Figure 6A*). We chose mNeonGreen as our fluorescent protein label because of its exceptional brightness, which aids detection at low levels of endogenous expression (*Shaner et al., 2013*). We sorted an mNeonGreen-expressing clonal cell line and confirmed proper labeling of EGFR with CD3ε-mNeonGreen by PCR of genomic DNA (*Figure 6B*) and immunoblotting of cell lysates (*Figure 6C*).

Based on our earlier results quantifying responses as a function of expression level (*Figure 1—figure supplements 2–4*), we reasoned that high-quality translocation would require low levels of ZtSH2 biosensor expression to match the expected lower levels of endogenous EGFR-ITAM expression produced by our CRISPR-tagging approach. We then transduced both parental HEK293T and EGFR-CD3ε-mNeonGreen knock-in cells with an mScarlet-labeled ZtSH2 to take advantage of this fluorescent protein's high brightness for detecting subcellular ZtSH2 localization even at low expression levels (*Bindels et al., 2017*). We then monitored ZtSH2 localization in low-mScarlet-expressing cells after treatment with EGF in both parental and knock-in cells. The resulting EGFR-pYtag knock-in HEK293T cells exhibited rapid clearance of ZtSH2 from the cytosol following EGF treatment, whereas no ZtSH2 response was observed in parental cells (*Figure 6D–F*, *Figure 6—video 1*). We therefore conclude that pYtags can be used to monitor the activation state of endogenously expressed RTKs, opening the door to single-cell studies of RTK signaling dynamics and endocytic trafficking in minimally perturbed contexts.

Surprisingly, we also observed striking differences in the dynamics of endogenous receptor activation compared to our prior experiments in which EGFR was overexpressed. For our EGFR-pYtag

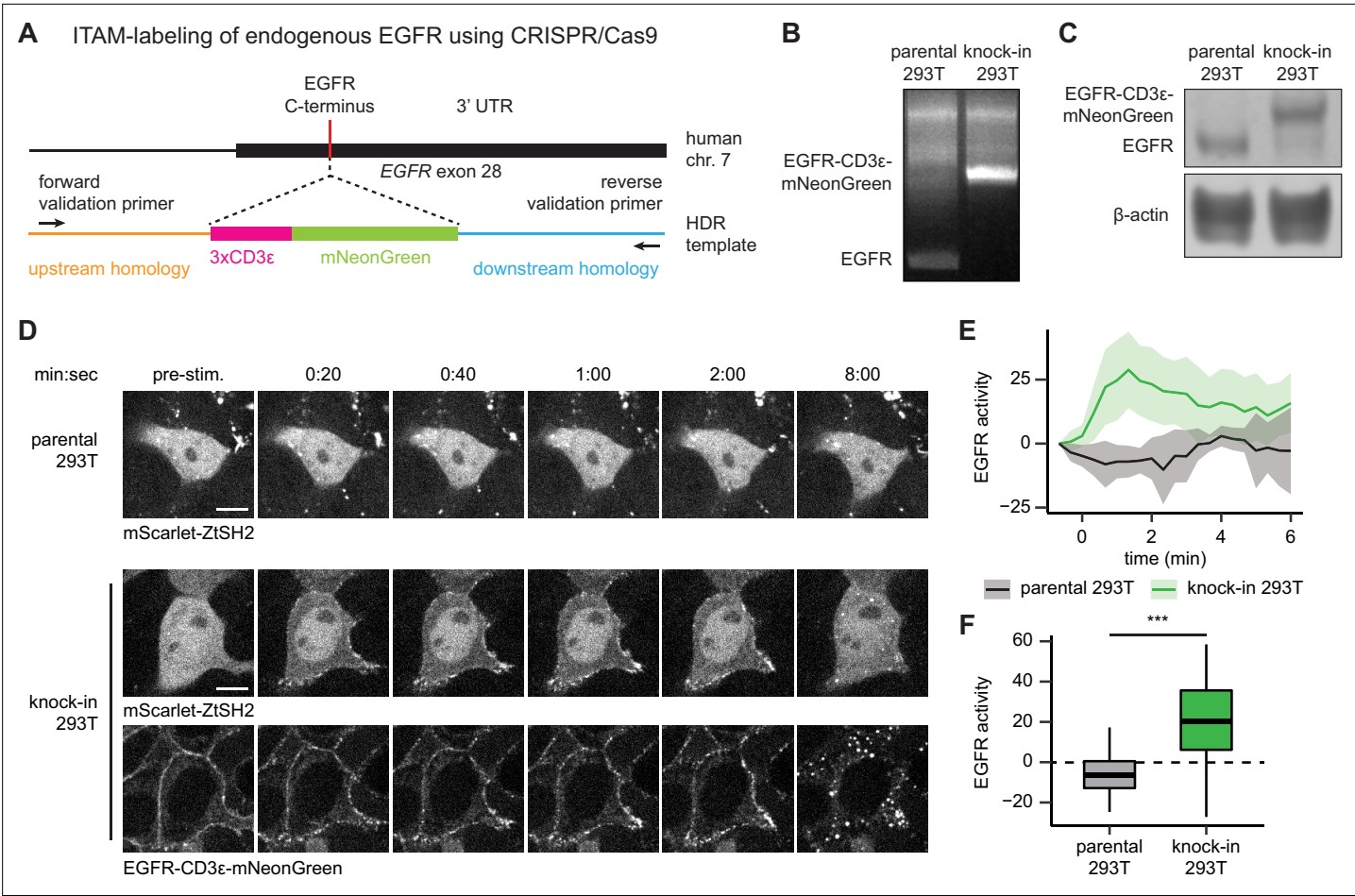

**Figure 6.** pYtags can be used to monitor the activity of endogenous receptor tyrosine kinases (RTKs). (**A**) Schematic of *EGFR* locus containing the C-terminus of epidermal growth factor receptor (EGFR), where CRISPR/Cas9 was used to label the receptor with CD3ε-mNeonGreen via homology-directed repair. (**B**) PCR of genomic DNA from parental or knock-in HEK293T cells. Validation primers targeting homology regions upstream and downstream of the CD3ε-mNeonGreen insert are labeled by black arrows in (**A**). (**C**) Immunoblots of EGFR in parental or knock-in HEK293T cells. (**D**) Images of parental or knock-in HEK293T cells treated with EGF (100 ng/mL). mScarlet-ZtSH2 images show averages of two successive frames to decrease background noise; full raw movie is included as *Figure 6—video 1*. Scale bar, 10 μm. (**E**) Mean ± SD clearance of ZtSH2 from the cytosol following treatment with EGF (100 ng/mL). parental HEK293T, n = 3 independent experiments; knock-in HEK293T, n = 4 independent experiments. (**F**) Clearance of ZtSH2 from the cytosol 1 min after treatment with EGF in (**E**). Lines denote mean values, boxes denote 25–75th percentiles, and whiskers denote minima and maxima. parental HEK293T, n = 23 cells from three independent experiments; knock-in HEK293T, n = 46 cells from four independent experiments. ***p<0.001 by Kolmogorov–Smirnov test. See also *Figure 6—video 1*.

The online version of this article includes the following video and source data for figure 6:

**Source data 1.** Uncropped gel for *Figure 6B*.

**Source data 2.** Uncropped gel for *Figure 6C*.

**Figure 6—video 1.** Timelapse of HEK293T cells expressing mScarlet-ZtSH2 and an endogenously labeled EGFR-CD3ε-mNeonGreen, treated with EGF (100 ng/mL).

https://elifesciences.org/articles/82863/figures#fig6video1

knock-in cells, we observed a decrease in ZtSH2 membrane localization within minutes after EGF stimulation, as well as near-complete loss of endogenous EGFR from the cell membrane within 8 min, with some internalized puncta that retained ZtSH2 labeling, likely reflecting internalization (*Figure 6D*, *Figure 6—video 1*). In comparison, HEK293T cells ectopically expressing high levels of an identical EGFR-pYtag system retained EGFR at the membrane for at least 10 min (*Figure 1—figure supplement 3A and B*). These data are in agreement with prior reports that EGFR endocytosis can be saturated at high levels of receptor expression (*Lund et al., 1990*) and suggest that the pYtag strategy

might be useful in future studies to relate EGFR activity, internalization, and trafficking at high resolution in single cells.

## Discussion

Here we describe pYtags, a biosensing strategy for monitoring the activity of a specific RTK in living cells. pYtags rely on tyrosine activation motifs that are selectively bound by corresponding tSH2 domains to minimize interactions with endogenous phosphotyrosine motifs and SH2 domains. This design principle confers selectivity to certain steps in T cell signaling, such as the recruitment of the kinase ZAP70 to phosphorylated ITAMs within activated TCRs. We exploit this property of immune signaling proteins to build two orthogonal pYtags: the first based on ITAMs from the TCR and the tSH2 domain of ZAP70, and the second based on the tyrosine-containing scaffold protein SLP76 and its SH2-containing binding partners Vav and ITK. We show that pYtags can be applied to several distinct RTKs (EGFR, ErbB2, FGFR1, PDGFRβ, and VEGFR3), providing a robust strategy to monitor receptor-level signaling in living cells.

pYtags quantitatively report on RTK activity with sub-minute temporal precision and can be applied to monitor signaling at both subcellular and multicellular length scales. Responses are highly reproducible across multiple experiments and independently derived cell lines, even when the same receptor is monitored using different pYtag variants (*Figure 5—figure supplement 1*). This quantitative precision enabled us to identify the differences in EGFR signaling dynamics induced by high- or low-affinity ligands (*Figure 3*), and to observe distinct signaling dynamics for two ErbB-family receptors (*Figure 5*). We used pYtag measurements to inform a mathematical model that, along with subsequent experimental validation, suggests that the dimerization affinity of ligand-bound receptors is important for specifying biphasic or mono-phasic EGFR signaling dynamics (*Figure 3*). Overall, our data suggest that pYtags represent a broadly applicable biosensing strategy to monitor the activity of any RTK of interest and can reveal new mechanistic insights for even the most well-studied RTKs.

pYtags have several advantages over previously reported biosensors of RTK activity. First, pYtags are modular: they can be applied to multiple RTKs without modification, unlike approaches that seek to identify phosphotyrosine/SH2 interactions present on endogenous RTKs (*Tiruthani et al., 2019*). Second, pYtags are specific, reporting only on the activity of the RTK labeled by the tyrosine activation motif. Third, pYtags can be multiplexed: these biosensors require one fewer fluorescent protein compared to FRET-based biosensors, and orthogonal variants of pYtags can be used to monitor distinct RTKs in the same cell. The human genome encodes at least 58 RTKs (*Lemmon and Schlessinger, 2010*), most of which remain poorly characterized with respect to their spatial organization and signaling dynamics. Multiplexing pYtags could open the door to systematically characterizing how different combinations of RTKs and cognate ligands influence signal processing, as has been performed for the bone morphogenetic protein receptor family (*Antebi et al., 2017*; *Klumpe et al., 2022*; *Su et al., 2022*).

Despite the advantages of pYtags, there are several challenges that might limit the adoption of this strategy. First, pYtags require a tyrosine activation motif to be appended to an RTK of interest, necessitating either ectopic expression of the labeled receptor or direct genome editing of the endogenous receptor. Although the increasingly powerful toolbox available for CRISPR-based genome editing lessens the burden associated with this strategy, endogenous tagging remains technically challenging and must be considered when engineering biosensor-expressing cell lines or organisms. A second limitation of pYtags is its reliance on phosphotyrosine/tSH2 motifs from immune signaling proteins. As a result, pYtags are currently restricted to nonimmune cells that do not endogenously express the corresponding proteins. It would be useful to further extend the approach to de novo designed substrate/binder pairs to broaden their applicability to additional cellular contexts.

We anticipate many future applications of the pYtag biosensing strategy. Live-cell biosensors of Erk signaling have revealed complex spatiotemporal signaling patterns – pulses, oscillations, and traveling waves – that appear to depend on the activation of certain RTKs (*Regot et al., 2014*; *De Simone et al., 2021*; *Hiratsuka et al., 2015*; *Pokrass et al., 2020*; *Simon et al., 2020*; *Hino et al., 2020*). Yet it remains unclear whether pulses of Erk activity require pulsatile activation of upstream RTKs. pYtags also open the door to quantitatively characterizing long-range ligand gradients that are thought to underlie processes such as collective cell migration or morphogen signaling during development (*Sprenger and Nüsslein-Volhard, 1992*; *Hino et al., 2020*). Finally, we expect that

the design principle of pYtags could be applied beyond RTK biosensors, such as for monitoring the signaling of non-receptor tyrosine kinases or engineering synthetic receptors with user-defined response programs.

## Methods

### Plasmid construction

All constructs were cloned into the pHR lentiviral expression plasmid using inFusion cloning. Linear DNA fragments were produced by PCR using HiFi polymerase (Takara Bio), followed by treatment with DpnI to remove template DNA. PCR products were then isolated through gel electrophoresis and purified using the Nucleospin gel purification kit (Takara Bio). Linear DNA fragments were then ligated using inFusion assembly and amplified in Stellar competent *Escherichia coli* (Takara Bio). Plasmids were purified by miniprep (QIAGEN) and verified by either Sanger sequencing (Genewiz) or whole-plasmid sequencing (Plasmidsaurus).

### Cell line generation

Constructs were stably expressed in cells using lentiviral transduction. First, lentivirus was produced by co-transfecting HEK293T LX cells with pCMV-dR8.91, pMD2.G, and the expression plasmid of interest. 48 hr later, viral supernatants were collected and passed through a 0.45 µm filter. Cells were seeded at ~30% confluency and transduced with lentivirus 24 hr later. 24 hr post-seeding, culture medium was replaced with medium containing 10 µg/mL polybrene and 200–300 µL viral supernatant was added to cells. Cells were then cultured in virus-containing medium for 48 hr. Populations of cells co-expressing each construct were isolated using fluorescence-activated cell sorting on a Sony SH800S cell sorter. Bulk-sorted populations were collected for all experiments, except for those using MCF10A cells and CRISPR/Cas9-edited HEK293T cells, for which clonal lines were generated. We validated the four cell lines used in this study (NIH3T3, HEK293T, MCF10A, and SYF MEFs) using STR profiling (ATCC Cell Authentication). Note that NIH3T3 (ATCC catalog CRL-1658) is distinct from 3T3-Swiss (ATCC catalog CCL-92), which may explain the differences in EGFR expression reported for these lines in various contexts (*Di Fiore et al., 1987*; *Livneh et al., 1986*; *Aharonov et al., 1978*).

### CRISPR/Cas9-based gene editing

HEK293T cells with CD3ε-mNeonGreen inserted at the endogenous *EGFR* locus were generated by transfecting cells with (1) a pX330 plasmid containing a human codon-optimized SpCas9 and guide RNA targeting *EGFR* (pX330 EGFR-sgRNA) and (2) a homology-directed repair template comprised of three repeats of the CD3ε ITAM and mNeonGreen flanked by 800 bp of *EGFR* homology regions (pUC19 EGFRup-CD3ε-mNeonGreen-EGFRdown) (*Ran et al., 2013*). HEK293T cells were first seeded at 130,000 cells/well in a 24-well plate. 24 hr later, cells were transfected with 330 ng pX330 EGFR-sgRNA and 170 ng pUC19 EGFRup-CD3ε-mNeonGreen-EGFRdown using Lipofectamine 3000 (Thermo Fisher Scientific) and left to culture for another 48 hr. Clonal cell lines were then isolated using fluorescence-activated cell sorting on a Sony SH800S cell sorter, and localization of mNeonGreen was assessed by confocal microscopy. A candidate clonal cell line exhibiting membrane-localized mNeon-Green was then used for further validation.

To verify genomic integration of CD3ε-mNeonGreen, genomic DNA was isolated from parental and knock-in HEK293T cells using a PureLink Genomic DNA Mini Kit (Invitrogen). PCR using HiFi polymerase was then used to amplify a region of genomic DNA using primers specific to homology regions flanking the CD3ε-mNeonGreen insertion (PEF122 and PEF123 in the Key Resources Table). PCR products were then run through an agarose gel by electrophoresis and imaged using an Axygen Gel Documentation System. In the gel, endogenous EGFR was expected to appear as a~1.2 kb product, while EGFR-CD3ε-mNeonGreen was expected to appear upshifted as an ~2.1 kb product. The expression of EGFR-CD3ε-mNeonGreen protein was verified by immunoblotting as described below.

### Cell culture

NIH3T3 cells, HEK293T cells, and SYF cells were cultured in DMEM (Gibco) supplemented with 10% fetal bovine serum (R&D Systems), 1% L-glutamine (Gibco), and 1% penicillin/streptomycin (Gibco).

MCF10A-5E cells (*Janes et al., 2010*) were cultured in DMEM/F12 (Gibco) supplemented with 5% horse serum (ATCC), 20 ng/mL EGF (R&D Systems), 0.5 µg/mL hydrocortisone (Corning), 100 ng/mL cholera toxin (Sigma-Aldrich), 10 µg/mL insulin (Sigma-Aldrich), and 1% penicillin/streptomycin (Gibco). All cells were maintained at 37°C and 5% $CO_2$. Cells were tested to confirm the absence of mycoplasma contamination.

## Polyacrylamide substrata

To prepare polyacrylamide substrata, 1.5-mm-thick glass coverslips were pretreated with glutaraldehyde. First, coverslips were treated with 0.1 N NaOH for 30 min, followed by rinsing with deionized water and air drying. Coverslips were then treated with 2% aminopropyltrimethoxysilane (Sigma-Aldrich) in acetone for 30 min, washed three times with acetone, and left to air dry. Coverslips were treated with 0.5% glutaraldehyde (Sigma-Aldrich) in PBS for 30 min, washed with deionized water, and left to air dry. Custom glass-bottom dishes were prepared by replacing the bottoms of 35 mm TCPS dishes with glutaraldehyde-treated coverslips, which were sealed using PDMS (Sigma-Aldrich).

Soft ($E \sim 0.1$ kPa) polyacrylamide substrata were made by first preparing a solution of 5% v/v acrylamide, 0.01% v/v bis-acrylamide, 0.05% v/v TEMED, and 0.05% v/v APS in deionized water. The acrylamide solution was pipetted onto a glass-bottom, glutaraldehyde-treated dish, sandwiched with an untreated coverslip, and allowed to gel for 1 hr at room temperature. The untreated coverslip was then removed, leaving a polyacrylamide hydrogel attached to the glutaraldehyde-treated coverslip. To coat polyacrylamide substrata with fibronectin, substrata were first washed with ethanol, washed three times with PBS, then washed once with HEPES buffer (50 mM, pH 8.5). 1 mg/mL sulfo-SANPAH (Thermo Fisher Scientific) in deionized water was pipetted onto the hydrogel, which was then subjected to UV crosslinking (2.8 J of 365 nm light exposure over 10 min). Substrata were then rinsed once with HEPES and treated again with sulfo-SANPAH and UV crosslinking. Substrata were rinsed three times with HEPES, coated with 100 µg/mL fibronectin (Corning) in PBS, and left at 4°C overnight before seeding cells the next day.

## Preparation of samples for live-cell imaging

For NIH3T3 and HEK293T experiments, cells were imaged on glass-bottom, black-walled 96-well plates (Cellvis) coated with fibronectin. Wells of 96-well plates were first incubated with 10 µg/mL fibronectin dissolved in PBS at 37°C for 30 min. Cells were seeded on glass-bottom 96-well plates at ~20,000 cells/well 1 d prior to imaging. 4 hr prior to imaging, the growth medium of cells was replaced with growth factor-free medium consisting of DMEM (Gibco) supplemented with 4.76 mg/mL HEPES and 3 mg/mL bovine serum albumin. Ligands were pre-diluted into growth factor-free medium, and 50 µL were added to a final well volume of 150 µL. These experimental parameters ensured an excess of ligand molecules to receptors, an appropriate regime for dose–response experiments to avoid ligand depletion from the medium (even assuming a high EGFR density of 100,000 receptors per cell and a low EGF concentration of 1 ng/mL, these experimental parameters would produce $2 \times 10^9$ EGFR receptors per well and $1.5 \times 10^{10}$ EGF ligands per well).

MCF10A cells were seeded on polyacrylamide substrata at ~400,000 cells/well 2 d prior to imaging. 24 hr prior to imaging, the growth medium of cells was replaced with growth factor-free medium consisting of DMEM/F12 (Gibco) supplemented with 0.5 µg/mL hydrocortisone, 100 ng/mL cholera toxin, 3 mg/mL bovine serum albumin, and 50 µg/mL penicillin/streptomycin. To prevent evaporation of media while imaging, 50 µL of mineral oil (VWR) was pipetted onto wells prior to mounting samples on the microscope.

## Live-cell imaging

Timelapse imaging of NIH3T3 cells, SYF cells, and HEK293T cells was performed on a Nikon Eclipse Ti microscope with a Yokogawa CSU-X1 spinning disk, an Agilent laser module containing 405, 488, 561, and 650 nm lasers, and an iXon DU897 EMCCD camera, using ×40 or ×60 oil objectives. Timelapse imaging of MCF10A cells on polyacrylamide substrata was performed on a Nikon Ti2-E microscope with a CSU-W1 SoRa spinning disk, a Hamamatsu FusionBT sCMOS camera, using a ×20 air objective with ×2.8 magnification optics.

## Immunoblotting

Cells were lysed in ice-cold RIPA buffer (1% Triton X-100, 50 mM HEPES, 150 mM NaCl, 1.5 mM MgCl$_2$, 1 mM EGTA, 100 mM NaF, 10 mM sodium pyrophosphate, 1 mM Na$_3$VO$_4$, 10% glycerol) supplemented with freshly prepared protease and phosphatase inhibitors. Protein levels were quantified using a Pierce BCA kit (Thermo Fisher Scientific), before being mixed with 6× Laemmli buffer/2-mercaptoethanol, heated for 5 min at 95°C, and loaded onto a 4–12% Bis-Tris gel (Invitrogen) for electrophoresis. Gels were transferred to a nitrocellulose membrane using the iBlot dry transfer system (Thermo Fisher Scientific), blocked in TBST with 5% milk for 30 min at room temperature and incubated in primary antibody overnight at 4°C. Before imaging, membranes were washed in TBST and treated with either IRDye 680CW or 800CW secondary antibodies (LI-COR) for 1 hr. Imaging was performed with the LI-COR Odyssey Infrared Imaging System. Immunoblot images were analyzed using FIJI. The signal of the target protein was normalized by the signal of β-actin, which was used as a loading control. To normalize EGFR phosphorylation levels (pEGFR) to those of total EGFR, β-actin-normalized pEGFR signals were divided by their respective β-actin-normalized EGFR signals from the same experiment.

## Immunostaining

Cells were fixed in 4% paraformaldehyde in PBS for 15 min, subjected to three 5 min washes with PBS, and incubated in blocking buffer (5% normal goat serum, 0.3% Triton X-100 in PBS) for 1 hr at room temperature. Samples were then incubated in antibody dilution buffer (2% bovine serum albumin, 0.3% Triton X-100 in PBS) containing primary antibodies overnight at 4°C. The following antibodies were used: anti-EEA1 (Cell Signaling Technology 48453); anti-ZAP70 (Cell Signaling Technology 3165). The next day, samples were subjected to three 5 min washes with PBS, incubated in antibody dilution buffer containing Alexa Fluor-conjugated secondary antibodies for 1 hr at room temperature, and subjected to another three 5 min washes with PBS. Samples were then imaged by confocal microscopy on a Nikon Eclipse Ti microscope as described above.

## Mathematical modeling
### System of ordinary differential equations (ODEs)

We used a system of ODEs to model the response of EGFR pYtag to soluble ligands, in cells initially residing in medium absent of EGFR-stimulating ligands. The model captures the following events using mass-action kinetics: (1) binding between ligands and receptors; (2) dimerization and autophosphorylation of receptors, which we treat as a single event; and (3) recruitment of ZtSH2 to ligand-bound EGFR dimers. Ligand binding, receptor dimerization/phosphorylation, and ZtSH2 recruitment were all treated as states that existed for individual receptors. We also assumed that soluble ligands in the cell culture medium were in vast excess to receptors, such that the concentration of soluble ligands is held constant. Since our simulations were run on relatively short time scales (~30 min post-stimulation), we did not consider trafficking and degradation of receptors. Recognizing that receptors could exist in every possible combination of these three states led to the following species and associated ODEs listed in *Appendix 1—table 1*.

### Parameters

We then parameterized our model using values reported previously. Rate constants and/or binding affinities for every step of the model were found, except for the joint dimerization/phosphorylation of receptors, which we estimated to qualitatively match the kinetics of pYtag responses observed in our experiments (*Appendix 1—table 2*).

## Initial conditions

Initial concentrations of EGFR-containing species ($N_3$-$N_{16}$) were determined for the case in which soluble ligand ($N_1$) is absent and no ZtSH2 is bound to receptors. In this case, the concentrations of ligand-bound and ZtSH2-bound EGFR ($N_4$; $N_6$-$N_{16}$) were set to zero. The concentration of total EGFR ($N_{E,o}$) was calculated by dividing the number of EGFR molecules per cell (assumed to be 250,000 per cell) *Herbst, 2004* by the occupied volume of the cell (treated as a sphere of 10 μm radius). Total EGFR and ZtSH2 were assumed to exist at a 1:1 molar ratio. In the absence of ligand, our model requires that EGFR is confined to the following two states: as a monomer ($N_3$) or as a ligand-free dimer

($N_5$). The initial concentrations of $N_3$ and $N_5$ were determined analytically. $N_{E,o}$ is the sum of the initial concentrations of monomeric EGFR ($N_{3,o}$) and EGFR in ligand-free dimers ($N_{5,o}$):

$$N_{E,o} = N_{3,o} + 2N_{5,o} \tag{1}$$

Setting the concentrations of all EGFR-containing species except for $N_{3,o}$ and $N_{5,o}$ to zero, Equations A3, A5, and A17 yield the following relationship between monomeric EGFR and its ligand-free dimer:

$$N_{5,o} = \frac{4N_{E,o} + \frac{k_6}{k_5} - \sqrt{\left(-4N_{E,o} - \frac{k_6}{k_5}\right)^2 - 16N_{E,o}^2}}{8} \tag{2}$$

$$N_{3,o} = N_{E,o} - 2N_{5,o} \tag{3}$$

## Model parameterization

Most parameters were set based on estimates from literature as detailed in *Appendix 1—table 2*. Two parameters ($k_5$ and $k_7$) represent forward and reverse rates for the lumped process of ligand-induced receptor dimerization and activation. These parameters were set manually to match the overall timescale and magnitude of ZtSH2 recruitment in response to EGF stimulation. To set the value of parameter $k_6$, which represents the dissociation rate of ligand-free, inactive EGFR dimers, we varied its value across two orders of magnitude and compared responses to the experimental timecourse in response to stimulation with 20 ng/mL EGF (*Figure 3—figure supplement 2*). We observed qualitative agreement for a value of $k_6 = 5 \times 10^{-3}$ s$^{-1}$ (*Figure 3—figure supplement 2D*).

## Tuning properties of EGFR ligands through β and γ

We assumed that different EGFR ligands may vary in their binding affinity for EGFR, which we modeled as an increase in the dissociation rate $k_7$ using a scaling parameter β. Different ligands have also been reported to induce different receptor conformations, leading to different dimerization affinities between ligand-bound receptors. We modeled this effect as an increase to the dissociation rate $k_3$ using a scaling parameter γ. To systematically test the effects of ligand–receptor binding and the dimerization affinity of ligand-bound receptors in *Figure 3C*, β and γ were increased by 1000- and 100-fold, respectively, and pYtag responses were simulated for ligand doses ranging from 0 to 5000 ng/mL.

## Simulating effects of GBM mutants

The dynamics of activation of WT EGFR and GBM-associated mutants were predicted in response to 20 ng/mL EREG. As described above, ligand–receptor binding affinity and ligand-bound dimerization affinity were decreased by increasing β and γ, respectively. To simulate WT EGFR exposed to EREG, we increased β 50-fold and increased γ 100-fold, relative to simulations of EGF treatment at the same ng/mL dose of ligand; the simulated response under these conditions was in qualitative agreement with the rapid, transient peak and transient plateau of signaling observed experimentally (*Figure 3A and E*). To simulate GBM-associated mutants treated with EREG, we increased β 6-fold and increased γ 650-fold (*Hu et al., 2022*), relative to WT EGFR treated with EREG.

## Quantification and statistical analysis

### Subtraction of background fluorescence from images

All images displayed in figures, and images used in subsequent analyses, were subjected to subtractions of background fluorescence either using a flat-field correction or by subtracting the intensity of cell-devoid regions from raw TIFF files. Images except for those of MCF10A cells on synthetic substrata were subjected to flat-field corrections. To perform a flat-field correction, raw TIFF files were imported into FIJI and subtracted of background fluorescence using a Gaussian-blurred image of a sample containing cell culture medium but lacking cells. For images of MCF10A cells on synthetic substrata, the mean gray value (intensity) of a region absent of cells was measured and subtracted from each pixel of the image at the same time point.

### Frame averaging for images of endogenous receptor activation

For the mScarlet-ZtSH2 images presented in *Figure 6D*, each image shows the average of two successive frames to reduce background noise. The full movie without averaging is also included as *Figure 6—video 1*.

### Quantification of pYtag and Grb2 biosensor responses

Cytosolic regions of randomly selected cells positive for both the fluorescently labeled RTK(s) and pYtag/Grb2 reporter(s) of interest were segmented in FIJI and the mean intensity was measured at each time point. In rare cases, abnormally dark or bright images were captured by the confocal microscope, causing sudden spikes in the cytosolic intensities measured in all cells; measurements from these aberrant images were rejected, and the cytosolic intensities of cells at the previous time point were used as a placeholder. CSV files containing cytosolic intensities of pYtag/Grb2 reporters were exported to R for subsequent analysis.

Raw cytosolic intensities of pYtag/Grb2 reporters were normalized to quantify the percentage of reporter cleared from the cytosol after stimulation with ligand. After validating each pYtag biosensor, the percentage of reporter cleared from the cytosol is hence referred to as the activity of the RTK of interest, $activity_{RTK}(t)$:

$$activity_{RTK}(t) = -100 * \left( \frac{I_{cyt}(t)}{I_{cyt,o}} - 1 \right) \tag{4}$$

where $I_{cyt}(t)$ is the cytosolic intensity of the reporter at a given time point, and $I_{cyt,o}$ is the mean cytosolic intensity of the reporter prior to stimulation with ligand. Mean pYtag/Grb2 responses over time were then calculated as the mean ± SD of population-averaged means from each experiment.

### Quantification of EGFR and EEA1 overlap in immunostained samples

Using 3D *z*-stack images of samples immunostained for EEA1, 3D masks of individual cells were segmented using CellPose, based on the fluorescent signal from FusionRed-labeled EGFR. 3D cell masks that spanned less than six continuous *z*-slices were discarded as these masks failed to capture the entire *z* dimension of the cells imaged. 3D cell masks were then subjected to erosion to remove the cell membrane from each mask. To isolate pixels positive for EEA1 or EGFR, EEA1 and EGFR images were subject to Otsu thresholding, and filtered for 3D objects larger than 64 pixels. Finally, the volume of pixels doubly positive for EGFR and EEA1 was quantified within each 3D cell mask using the corresponding thresholded images of EGFR and EEA1.

### Quantification of EGFR membrane intensity in immunostained samples

Using 3D *z*-stack images of FusionRed-labeled EGFR, *z*-slices capturing the mid-plane of cells were first selected in FIJI. For each cell, the Straight Line feature (width, 10 pixels) was used to draw a line perpendicular to the cell membrane, and a line scan of EGFR fluorescence was measured. From this line scan, the maximum fluorescence value was defined as the intensity of EGFR at the cell membrane.

### Quantification of ZtSH2 and EGFR membrane localization in MCF10A cells

Enrichment of ZtSH2 and EGFR at cell membranes in *Figure 2* was quantified in FIJI. The Straight Line feature was used to draw a region of interest intersecting perpendicularly with either media-exposed membranes or cell–cell contacts. Intensity profiles for ZtSH2 and EGFR channels were then measured using the Plot Profile feature and exported to R for analysis.

### Quantification of pYtag and ErkKTR responses in MCF10A cells

Individual *z*-slices from 3D timelapse imaging were used to quantify both EGFR pYtag and ErkKTR responses in MCF10A cells cultured on soft substrata. EGFR pYtag responses were quantified as described above (see 'Quantification of pYtag and Grb2 biosensor responses'). Using the ErkKTR channel, nuclear and cytoplasmic regions of individual cells were segmented in FIJI and the intensity

of these regions was measured at each time point. ErkKTR-reported Erk activity was calculated by dividing the cytosolic KTR intensity by the nuclear KTR intensity at each time point. Each cell's EGFR pYtag and ErkKTR trajectories were normalized to their respective minimum and maximum readouts for the reporter of interest.

## Statistical analysis and replicates

All experiments were performed over at least three biological replicates or two independent experiments. Biological replicates are defined as biologically distinct samples aimed to capture biological variation. Independent experiments are defined as biologically distinct samples prepared and analyzed on separate days.

## Resource details

### Lead contact

Further information and requests for resources and reagents should be directed and will be fulfilled by the lead contact, Jared Toettcher (toettcher@princeton.edu).

### Materials availability

There are no restrictions on material availability. Plasmids are available from Addgene (https://www.addgene.org/Jared_Toettcher); all cell lines produced and plasmids unavailable on Addgene will be made available upon request.

## Acknowledgements

We thank all members of the Bashor, Nelson, and Toettcher labs for their insights and comments, and Gary Laevsky and Sha Wang from the Princeton Nikon Imaging Facility for assistance with microscopy. This work was supported by NSF CAREER Award 1750663 and a Vallee Scholar award (to JET); NIH grants R01HL164861, R01HD099030, and DP1HD111539 (to CMN); a Schmidt Transformative Technology Award (to CMN and JET); NIH training grant T32GM007388 (to EVM); NIH grant R01EB032272 and ONR grant N00014-21-1-4006 (to CJB). PEF was supported in part by the NSF Graduate Research Fellowship Program. EHB-R was supported by the National Institute of Natural Sciences, Japan.

## Additional information

### Funding

| Funder | Grant reference number | Author |
| --- | --- | --- |
| National Science Foundation | 1750663 | Jared E Toettcher |
| Vallee Foundation | | Jared E Toettcher |
| National Institutes of Health | R01HL164861 | Celeste M Nelson |
| National Institutes of Health | R01HD0099030 | Celeste M Nelson |
| National Institutes of Health | DP1HD111539 | Celeste M Nelson |
| National Institutes of Health | T32GM007388 | Emily V Mesev |
| Office of Naval Research | N00014-21-1-4006 | Caleb J Bashor |
| National Science Foundation | GRFP | Payam E Farahani |
| National Institutes of Health | R01EB032272 | Caleb J Bashor |

| Funder | Grant reference number | Author |
| --- | --- | --- |
| National Science Foundation | 2134935 | Celeste M Nelson<br>Jared E Toettcher |

The funders had no role in study design, data collection and interpretation, or the decision to submit the work for publication.

## Author contributions

Payam E Farahani, Conceptualization, Resources, Software, Investigation, Methodology, Writing – original draft, Writing – review and editing; Xiaoyu Yang, Conceptualization, Methodology; Emily V Mesev, Kaylan A Fomby, Ellen H Brumbaugh-Reed, Investigation; Caleb J Bashor, Conceptualization, Supervision, Funding acquisition, Writing – review and editing; Celeste M Nelson, Jared E Toettcher, Conceptualization, Supervision, Funding acquisition, Writing – original draft, Writing – review and editing

## Author ORCIDs

Payam E Farahani ⬥ http://orcid.org/0000-0003-2912-0519
Xiaoyu Yang ⬥ http://orcid.org/0000-0001-5636-117X
Caleb J Bashor ⬥ http://orcid.org/0000-0003-1354-2098
Celeste M Nelson ⬥ http://orcid.org/0000-0001-9973-8870
Jared E Toettcher ⬥ http://orcid.org/0000-0002-1546-4030

## Decision letter and Author response

Decision letter https://doi.org/10.7554/eLife.82863.sa1
Author response https://doi.org/10.7554/eLife.82863.sa2

---

# Additional files

## Supplementary files

• MDAR checklist

## Data availability

There are no restrictions on material availability. Plasmids are available from Addgene (https://www.addgene.org/Jared_Toettcher); all cell lines produced and plasmids unavailable on Addgene will be made available upon request. There are no restrictions on data availability. Source data generated or analyzed during this study, as well as Python and R scripts for data analysis and mathematical modeling, are available on the laboratory GitHub page (https://github.com/toettchlab/Farahani2022/; copy archived at *toettchlab, 2023*).

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

# Appendix 1

## Appendix 1—key resources table

| Reagent type (species) or resource | Designation | Source or reference | Identifiers | Additional information |
|---|---|---|---|---|
| Cell line (human) | MCF10A-5E | *Janes et al., 2010* | RRID:CVCL_0598 | |
| Cell line (human) | HEK293T LX | ClonTech Laboratories | Cat # 632180 | |
| Cell line (mouse) | NIH3T3 | ATCC | Cat # CRL-1658 | |
| Cell line (mouse) | SYF mouse embryonic fibroblasts (MEFs) | ATCC | Cat # CRL-2459 | |
| Cell line (*Escherichia coli*) | Stellar chemically competent cells | ClonTech Laboratories | Cat # 636763 | |
| Recombinant DNA reagent | pCMV-dR8.91 lentivirus packaging plasmid | Gift from Prof. Didier Trono, EPFL | Addgene # 12263 | |
| Recombinant DNA reagent | pMD2.G lenti helper plasmid | Gift from Prof. Didier Trono, EPFL | Addgene # 12259 | |
| Recombinant DNA reagent | pHR EGFR-FusionRed | *Yang et al., 2021* | Addgene # 179263 | |
| Recombinant DNA reagent | pHR EGFR-CD3 ζ 1-FusionRed | This paper | N/A | EGFR-ITAM construct |
| Recombinant DNA reagent | pHR EGFR-CD3 ζ 2-FusionRed | This paper | N/A | EGFR-ITAM construct |
| Recombinant DNA reagent | pHR EGFR-CD3 ζ 3-FusionRed | This paper | N/A | EGFR-ITAM construct |
| Recombinant DNA reagent | pHR EGFR-CD3γ-FusionRed | This paper | N/A | EGFR-ITAM construct |
| Recombinant DNA reagent | pHR EGFR-CD3δ-FusionRed | This paper | N/A | EGFR-ITAM construct |
| Recombinant DNA reagent | pHR EGFR-CD3ε-FusionRed | This paper | Addgene # 188626 | EGFR-ITAM construct |
| Recombinant DNA reagent | pHR EGFR- CD3ε-mNeonGreen | This paper | N/A | EGFR-ITAM construct |
| Recombinant DNA reagent | pHR iRFP-ZtSH2 | This paper | Addgene # 188627 | ZtSH2 biosensor |
| Recombinant DNA reagent | pHR mScarlet-ZtSH2 | This paper | N/A | ZtSH2 biosensor |
| Recombinant DNA reagent | pHR EGFR(R84K)-CD3ε-FusionRed | This paper | N/A | GBM-mutant EGFR construct (*Figure 3*) |
| Recombinant DNA reagent | pHR EGFR(A265V)-CD3ε-FusionRed | This paper | N/A | GBM-mutant EGFR construct (*Figure 3*) |
| Recombinant DNA reagent | pHR ErbB2-FusionRed | This paper | N/A | ITAM-less ErbB2 construct (*Figure 4*) |
| Recombinant DNA reagent | pHR ErbB2- CD3ε-FusionRed | This paper | Addgene # 188628 | ITAM-tagged ErbB2 (*Figure 4*) |
| Recombinant DNA reagent | pHR EGFR-Citrine | This paper | N/A | Fluorescent EGFR construct (*Figure 4*) |
| Recombinant DNA reagent | pHR Clover-VISH2 | This paper | Addgene # 188629 | tSH2 biosensor (*Figure 5*) |
| Recombinant DNA reagent | pHR EGFR-CD3ε-TagBFP | This paper | N/A | ITAM-tagged EGFR (*Figure 5*) |

*Appendix 1 Continued on next page*

*Appendix 1 Continued*

| Reagent type (species) or resource | Designation | Source or reference | Identifiers | Additional information |
|---|---|---|---|---|
| Recombinant DNA reagent | pHR EGFR-SLP76-TagBFP | This paper | Addgene # 188630 | ITAM-tagged EGFR (*Figure 5*) |
| Recombinant DNA reagent | pHR Grb2-TagBFP | This paper | Addgene # 188631 | Grb2-based biosensor (*Figure 1*) |
| Recombinant DNA reagent | pHR FGFR1-CD3ε-FusionRed | This paper | Addgene # 188632 | ITAM-tagged FGFR1 (*Figure 4*) |
| Recombinant DNA reagent | pHR FGFR1-FusionRed | This paper | N/A | ITAM-less FGFR1 (*Figure 4*) |
| Recombinant DNA reagent | pHR PDGFRβ-CD3ε-FusionRed | This paper | N/A | ITAM-tagged PDGFR (*Figure 4*) |
| Recombinant DNA reagent | pHR PDGFRβ-FusionRed | This paper | N/A | ITAM-less PDGFR (*Figure 4*) |
| Recombinant DNA reagent | pHR VEGFR3-CD3ε-FusionRed | This paper | N/A | ITAM-tagged VEGFR (*Figure 4*) |
| Recombinant DNA reagent | pHR VEGFR3-FusionRed | This paper | N/A | ITAM-less VEGFR (*Figure 4*) |
| Recombinant DNA reagent | pHR ErkKTR-TagBFP | 9 | N/A | |
| Recombinant DNA reagent | pX330 EGFR-sgRNA | This paper, using a plasmid from Feng Zhang, MIT | Addgene # 188633 | EGFR-targeting gRNA (*Figure 6*) |
| Recombinant DNA reagent | pUC19 EGFRup-CD3ε-mNeonGreen-EGFRdown | This paper | Addgene # 188634 | CRISPR plasmid for EGFR modification (*Figure 6*) |
| Sequence-based reagent | PEF122 forward primer | This paper | 5'- TTCTTTTGCAGCAAC AGCAAGAGGGCCCTCCC-3' | Used to verify CRISPR tagging; see 'Methods' |
| Sequence-based reagent | PEF123 reverse primer | This paper | 5'- TCCGTTTCTTCTTTGCCCAG GAAGGGACAGAGTGGCTTATCC-3' | Used to verify CRISPR tagging; see 'Methods' |
| Antibody | Anti-EGFR antibody (rabbit monoclonal) | Cell Signaling Technology | Cat # 4267 | Used at 1:1000 for western blotting |
| Antibody | Anti-pEGFR antibody (rabbit monoclonal) | Cell Signaling Technology | Cat # 3777 | Used at 1:1000 for western blotting |
| Antibody | Anti-β-actin antibody (mouse monoclonal) | Cell Signaling Technology | Cat # 3700 | Used at 1:1000 for western blotting |
| Antibody | Anti-pAkt antibody (rabbit polyclonal) | Cell Signaling Technology | Cat # 9271 | Used at 1:1000 for western blotting |
| Antibody | Anti-ppErk antibody (rabbit polyclonal) | Cell Signaling Technology | Cat # 9101 | Used at 1:1000 for western blotting |
| Antibody | Anti-EEA1 antibody (mouse monoclonal) | Cell Signaling Technology | Cat # 48453 | Used at 1:100 for immunostaining |
| Antibody | Anti-ZAP70 antibody (rabbit monoclonal) | Cell Signaling Technology | Cat # 3165 | Used at 1:1000 for western blotting |
| Antibody | Goat anti-Mouse IgG (H+L) Cross-Adsorbed Secondary Antibody, Alexa Fluor 488 (goat polyclonal) | Invitrogen | Cat # A-11001 | Used at 1:500 for immunostaining |

*Appendix 1 Continued on next page*

*Appendix 1 Continued*

| Reagent type (species) or resource | Designation | Source or reference | Identifiers | Additional information |
|---|---|---|---|---|
| Antibody | Goat anti-Rabbit IgG (Heavy chain), Superclonal Recombinant Secondary Antibody, Alexa Fluor 647 (goat polyclonal) | Invitrogen | Cat # A27040 | Used at 1:500 for immunostaining |
| Antibody | Goat anti-Mouse IgG (H+L) Highly Cross-Adsorbed Secondary Antibody, Alexa Fluor 647 (goat polyclonal) | Invitrogen | Cat # A-21236 | Used at 1:500 for immunostaining |
| Antibody | IRDye 680RD Goat anti-Mouse IgG antibody (goat polyclonal) | LI-COR | Cat # 926-68070 | Used at 1:10,000 for western blotting |
| Antibody | IRDye 800CW Goat anti-Rabbit IgG antibody (goat polyclonal) | LI-COR | Cat # 926-32211 | Used at 1:10,000 for western blotting |
| Peptide, recombinant protein | Bovine serum albumin | Sigma-Aldrich | Cat # 12659 | |
| Peptide, recombinant protein | Fibronectin | Corning | Cat # CB-40008A | Cell adhesion coating |
| Peptide, recombinant protein | ClonAmp HiFi PCR polymerase | ClonTech Laboratories | Cat # 639298 | Polymerase |
| Peptide, recombinant protein | Insulin | Sigma-Aldrich | Cat # I6634 | |
| Peptide, recombinant protein | Cholera toxin | Sigma-Aldrich | Cat # C8052 | |
| Peptide, recombinant protein | L-glutamine | Gibco | Cat # 25030-081 | |
| Peptide, recombinant protein | EGF | R&D Systems | Cat # 236-EG-200 | |
| Peptide, recombinant protein | Epiregulin | R&D Systems | Cat # 1195-EP-025 | |
| Peptide, recombinant protein | Epigen | R&D Systems | Cat # 6629-EP-025 | |
| Peptide, recombinant protein | FGF4 | R&D Systems | Cat # 235-F4-025 | |
| Peptide, recombinant protein | PDGF-BB | Millipore Sigma | Cat # P3201 | |
| Peptide, recombinant protein | VEGF-C | R&D Systems | Cat # 9199-VC-025 | |
| Chemical compound, drug | Gefitinib | Cell Signaling Technology | Cat # 4765 | |
| Chemical compound, drug | Hydrocortisone | Sigma-Aldrich | Cat # H0888 | |

*Appendix 1 Continued on next page*

*Appendix 1 Continued*

| Reagent type (species) or resource | Designation | Source or reference | Identifiers | Additional information |
|---|---|---|---|---|
| Chemical compound, drug | Penicillin/ streptomycin | Gibco | Cat # 15140–122 | |
| Chemical compound, drug | TrypLE Express | Gibco | Cat # 12605-028 | |
| Chemical compound, drug | FuGENE HD | Promega | Cat # E2311 | |
| Chemical compound, drug | Lipofectamine 3000 | Thermo Fisher Scientific | Cat # L3000015 | |
| Chemical compound, drug | Aminopropyl trimethoxysilane | Sigma-Aldrich | Cat # 281778 | |
| Chemical compound, drug | Glutaraldehyde | Sigma-Aldrich | Cat # 340855 | |
| Chemical compound, drug | 40% acrylamide solution | Bio-Rad | Cat # 1610140 | |
| Chemical compound, drug | 2% bis-acrylamide solution | Bio-Rad | Cat # 161-0142 | |
| Chemical compound, drug | N,N,N',N'-Tetramethyl ethylenediamine (TEMED) | Sigma-Aldrich | Cat # T9281 | |
| Chemical compound, drug | Ammonium persulfate (APS) | Sigma-Aldrich | Cat # A3678 | |
| Commercial assay or kit | inFusion HD cloning kit | ClonTech Laboratories | Cat # 638911 | Cloning kit |
| Other | DMEM/F12 | Gibco | Cat # 11320033 | Culture media |
| Other | Horse serum | Gibco | Cat # 16050122 | Serum for culture media |
| Other | DMEM | Gibco | Cat # 11995-065 | Culture media |
| Other | Fetal bovine serum | R&D Systems | Cat # S11150 | Serum for culture media |
| Software, algorithm | FIJI | *Schindelin et al., 2012* | http://fiji.sc; RRID:SCR_00228 | |
| Software, algorithm | Python code for computational model; analysis code for raw data | This paper | https://github.com/toettchlab/Farahani2022/ (copy archived at *toettchlab, 2023*) | |
| Software, algorithm | R Studio 1.1.456 | RStudio | rstudio.com; RRID:SCR_000432 | |

**Appendix 1—table 1.** Equations used in the mathematical model.
L-EGFR: ligand-bound EGFR; EGFR:EGFR: EGFR in dimeric form; EGFR: EGFR bound to ZtSH2.

| Species | Notation | Equation |
|---|---|---|
| Soluble ligand (L) | $N_1$ | $\frac{dN_1}{dt} = 0$ (A1) |
| Unbound ZtSH2 (*) | $N_2$ | $\frac{dN_2}{dt} = -k_8 N_2 * (2N_6 + 2N_7 + N_9 + N_{10} + N_{11}) + k_9 * (N_8 + N_9 + N_{10} + N_{11} + 2N_{12} + 2N_{13} + 2N_{14} + N_{15} + N_{16})$ (A2) |
| EGFR | $N_3$ | $\frac{dN_3}{dt} = -k_1 N_3 N_1 + k_2 N_4 - 2k_5 N_3^2 - k_5 N_3 N_4 - k_5 N_3 N_{15} - k_5 N_3 N_{16} + 2k_6 N_5 + k_6 N_8 + k_7 N_6 + k_7 N_9 + k_9 N_{15}$ (A3) |
| L-EGFR | $N_4$ | $\frac{dN_4}{dt} = k_1 N_3 N_1 - k_2 N_4 - k_5 N_3 N_4 - 2k_5 N_4^2 - k_5 N_4 N_{15} - k_5 N_4 N_{16} + k_7 N_6 + k_7 N_{10} + 2k_7 N_7 + k_7 N_{11} + k_9 N_{16}$ (A4) |
| EGFR:EGFR | $N_5$ | $\frac{dN_5}{dt} = -2k_1 N_5 N_1 + k_3 N_6 + k_5 N_3^2 - k_6 N_5 + k_9 N_8$ (A5) |
| L-EGFR:EGFR | $N_6$ | $\frac{dN_6}{dt} = -k_1 N_6 N_1 + 2k_4 N_7 + 2k_1 N_5 N_1 - k_3 N_6 + k_5 N_3 N_4 - k_7 N_6 - 2k_8 N_6 N_2 + k_9 N_9 + k_9 N_{10}$ (A6) |

*Appendix 1—table 1 Continued on next page*

*Appendix 1—table 1 Continued*

| Species | Notation | Equation |
|---|---|---|
| L-EGFR:L-EGFR | $N_7$ | $\frac{dN_7}{dt} = k_5 N_6 N_1 - 2k_4 N_7 + k_5 N_4^2 - k_7 N_7 - 2k_8 N_7 N_2 + k_9 N_{11}$ (A7) |
| EGFR\*:EGFR | $N_8$ | $\frac{dN_8}{dt} = -2k_1 N_8 N_1 + k_3 N_9 + k_3 N_{10} + k_5 N_3 N_{15} - k_6 N_8 - k_9 N_8 + 2k_9 N_{12}$ (A8) |
| L-EGFR\*:EGFR | $N_9$ | $\frac{dN_9}{dt} = k_1 N_8 N_1 - k_1 N_9 N_1 - k_3 N_9 + k_4 N_{11} + k_5 N_3 N_{16} - k_7 N_9 + k_8 N_6 N_2 - k_8 N_9 N_2 - k_9 N_9 + k_9 N_{13}$ (A9) |
| L-EGFR:EGFR\* | $N_{10}$ | $\frac{dN_{10}}{dt} = k_1 N_8 N_1 - k_3 N_{10} - k_1 N_{10} N_1 + k_4 N_{11} + k_5 N_4 N_{15} - k_7 N_{10} + k_8 N_6 N_2 - k_9 N_{10} - k_8 N_{10} N_2 + k_9 N_{13}$ (A10) |
| L-EGFR\*:L-EGFR | $N_{11}$ | $\frac{dN_{11}}{dt} = k_1 N_9 N_1 - 2k_4 N_{11} + k_1 N_{10} N_1 + k_5 N_4 N_{16} - k_7 N_{11} + 2k_8 N_7 N_2 - k_9 N_{11} - k_8 N_{11} N_2 + 2k_9 N_{14}$ (A11) |
| EGFR\*:EGFR\* | $N_{12}$ | $\frac{dN_{12}}{dt} = -2k_1 N_{12} N_1 + k_3 N_{13} + k_5 N_{15}^2 - k_6 N_{12} - 2k_9 N_{12}$ (A12) |
| L-EGFR\*:EGFR\* | $N_{13}$ | $\frac{dN_{13}}{dt} = 2k_1 N_{12} N_1 - k_3 N_{13} - k_1 N_{13} N_1 + 2k_4 N_{14} + k_5 N_{15} N_{16} - k_7 N_{13} + k_8 N_9 N_2 - 2k_9 N_{13} + k_8 N_{10} N_2$ (A13) |
| L-EGFR\*:L-EGFR\* | $N_{14}$ | $\frac{dN_{14}}{dt} = k_1 N_{13} N_1 - 2k_4 N_{14} + k_5 N_{16}^2 - k_7 N_{14} + k_8 N_{11} N_2 - 2k_9 N_{14}$ (A14) |
| EGFR\* | $N_{15}$ | $\frac{dN_{15}}{dt} = -k_1 N_{15} N_1 + k_2 N_{16} - 2k_5 N_{15}^2 + 2k_6 N_{12} - k_5 N_3 N_{15} + k_6 N_8 - k_5 N_{15} N_{16} + k_7 N_{13} - k_5 N_4 N_{15} + k_7 N_{10} - k_9 N_{15}$ (A15) |
| L-EGFR\* | $N_{16}$ | $\frac{dN_{16}}{dt} = k_1 N_{15} N_1 - k_2 N_{16} - 2k_5 N_{16}^2 + 2k_7 N_{14} - k_5 N_{15} N_{16} + k_7 N_{13} - k_5 N_{16} N_3 + k_7 N_9 - k_5 N_4 N_{16} + k_7 N_{11} - k_9 N_{16}$ (A16) |

**Appendix 1—table 2.** Parameters used in the mathematical model.

| Parameter | Notation | Value | Units | Notes |
|---|---|---|---|---|
| Receptor–ligand binding | $k_1$ | 0.03 | nM$^{-1}$ s$^{-1}$ | *Schoeberl et al., 2002* |
| Ligand dissociating from L-EGFR ($\beta$ = 1 corresponds to EGF) | $k_2$ | $\beta$*6.6e-3 | s$^{-1}$ | *Macdonald and Pike, 2008* |
| Ligand dissociating from L-EGFR:EGFR ($\beta$ = 1 corresponds to EGF) | $k_3$ | $\beta$*5.7e-3 | s$^{-1}$ | *Macdonald and Pike, 2008* |
| Ligand dissociating from L-EGFR:L-EGFR | $k_4$ | 0.087 | s$^{-1}$ | *Macdonald and Pike, 2008* |
| Receptor dimerization and activation | $k_5$ | 1e-5 | nM$^{-1}$ s$^{-1}$ | Estimated |
| EGFR:EGFR dissociation | $k_6$ | 5e-3; variable values in *Figure 3—figure supplement 2*. | s$^{-1}$ | Estimated |
| L-EGFR:EGFR dissociation ($\gamma$ = 1 corresponds to EGF) | $k_7$ | $\gamma$*1e-4 | s$^{-1}$ | Estimated |
| ZtSH2 binding to receptor | $k_8$ | 5 | nM$^{-1}$ s$^{-1}$ | Kd from *Ottinger et al., 1998*; kinetics set to be ~10 s based on our experimental measurements of ZtSH2 translocation |
| ZtSH2 dissociating from receptor | $k_9$ | 16.67 | s$^{-1}$ | Kd from *Ottinger et al., 1998*; kinetics set to be ~10 s based on our experimental measurements of ZtSH2 translocation |

*Appendix 1—table 2 Continued on next page*

*Appendix 1—table 2 Continued*

| Parameter | Notation | Value | Units | Notes |
|-----------|----------|-------|-------|-------|
| Scaling parameter for ligand–receptor binding | β | 1 for EGF; 50 for low-affinity ligands; variable values in *Figure 3C* | Unitless | *Freed et al., 2017* |
| Scaling parameter for dimerization of ligand-bound receptors | γ | 1 for EGF; 100 for low-affinity ligands; variable values in *Figure 3C* | Unitless | *Freed et al., 2017*; *Hu et al., 2022* |

