## [Editor Report]

Your study provides strong and convincing evidence that pYtags enable spatiotemporal measurements of receptor tyrosine kinase signaling in living cells. This is highly significant as it can be used to study in real-time receptor signaling in healthy and diseased cells.

---

## [Decision Letter]

**Decision letter after peer review:**

Thank you for submitting your article "pYtags enable spatiotemporal measurements of receptor tyrosine kinase signaling in living cells" for consideration by *eLife*. Your article has been reviewed by 3 peer reviewers, and the evaluation has been overseen by a Reviewing Editor and Jonathan Cooper as the Senior Editor. The following individual involved in the review of your submission has agreed to reveal their identity: Alex J B Kreutzberger (Reviewer #3).

Overall, the reviewers found your new approach, using RTK-ITAM fusions and fluorescent Syk or Zap70 SH2 domains to monitor RTK activation, to be exciting and innovative. The approach is explained well, and the experiments are clear and clean. However, as you will see from the original reviews and comments below, there was a divergence in opinion on the interpretation of the data and thus on the likely adoption of this new tool by the research community. During the review consultation, several themes emerged:

1. Use of different cell types makes it difficult to understand the effect of expression level (of RTK fusion and of SH2 reporter) on the kinetics of probe depletion from the cytosol. There are indications that the assay is capable of being very sensitive, but the use of multiple cell types, expression levels and measurement times makes it very difficult to determine the caveats of the system and the types of applications for which this is most useful. We recommend CRISPR/Cas9 insertion of the ITAM tag into the EGFR gene in the same cell type that is used for over-expression. NIH3T3 cells would perhaps make most sense, since they endogenously express ~40K EGFRs, or MCF10A, where many EGFR kinetics and imaging studies have been done.

2. Discrepancy between apparent sensitivity of the approach (ability to detect EGFR activation using the knock-in approach in HEK 293 cells, in which EGFR expression is thought to be very low), with the inability to detect ERBB2 activation by EGF unless EGFR is co-over-expressed. Use of supra-physiological concentrations of ligand (EGF) is also a concern. The sensitivity of the system should be tested with lower EGF concentrations.

3. Lack of calibration of the experiments. The readout of the system (% clearance) is highly dependent on the ratio of tagged receptor to fluorescent probe, the rate of access of the probe, and the number of internal pools and their rate of exchange. RTK activation will also depend on the ratio of the ligand to receptor as well as ligand affinity. However, it is felt that these parameters have not been properly taken into account or controlled.

4. The mathematical model makes use of seemingly arbitrary calibration factors rather than using biophysical quantities that have been previously established for the EGFR system. One reviewer questioned whether the mathematical model for dimerization is indeed needed and whether it enhances the message. If the authors feel that the mathematical model should be included, then the authors should consider cleanly calibrating an experiment to better parameterize the math model. If the math model could be used to extract fundamental numbers for the EGFR system, then the same principles could be applied to other ligand-receptor systems analyzed by the pYtag approach.

5. Since endocytosis of the tagged EGFR seems slower than reported for endogenous EGFRs in this system, controls are needed to show that tagging does not perturb EGFR signaling or internalization. Some type of EGF uptake or degradation assay in the cells with endogenous EGFR plus/min the tagging system would be an appropriate control.

Overall, it is felt that, as written, the manuscript provides a proof-of-concept rather than making a strong case for others to adapt this assay to study their receptors of interest.

*Reviewer #1 (Recommendations for the authors):*

1. On page 9, the authors state: "Notably, we observed that pYtag-expressing cells stimulated for at least 30 min with EGF contained internalized vesicles that were positive for both total EGFR and ZtSH2 (Figures 2C and 2D). Subsequent treatment with Gefitinib eliminated ZtSH2 from EGFR-positive vesicles within minutes, suggesting that the enrichment of ZtSH2 at vesicles is indicative of signaling from endosomal compartments (Figures 2C and 2D)." To conclude that the accumulation of EGFR and ZtSH2 in punctae that are vesicles/endosomal compartments simultaneous labeling with appropriate markers should be performed (see also below, point 2). The same applies to experiments performed on CRISPR-Cas9 genome-edited HEK 293T cells [page 18: "We also observed rapid and near-complete internalization of endogenous EGFR from the cell membrane, with some internalized vesicles retaining residual ZtSH2 labeling (Figure 6D, right-most panels)."].

2. It would be interesting (and maybe useful to add some novel insights) to follow the receptor routes after it has reached the cell membrane to observe whether the activation also occurs in (or from) specific endocytic compartments (e.g. Rab5 or EEA-1 positive early endosomes, Rab7 or LAMP1 positive late endosomes/lysosomes, Rab11 positive perinuclear/recycling compartment). This set of experiments could be performed by either simultaneously expressing fluorescent protein-tagged endosomal/vesicular markers or staining the corresponding endogenous proteins in cells that were fixed at different time points upon ligand stimulation.

3. The experiments over-expressing the pYtags were mainly performed in NIH3T3 cells, whereas the CRISPR-Cas9 knock-in was done in HEK 293T. It would be useful to check whether the same differences would be also seen in HEK 293T over-expressing the tag. After assessing this, further characterization of the signal emitted by the cell line generated by CRISPR-Cas9 would add some key concepts to RTK activity dynamics (e.g. the localization of the RTK activation signal in the genetically manipulated cells in Figure 6, looks quite different from the signal shown in Figure 1. In Figure 6, it looks more dot-like and less localized in cell-to-cell contacts. Do the authors have an explanation for this?).

4. Some experiments shown in Figures 1 and 2 were repeated only two times. The analysis of at least three independent experiments is in general a well-accepted standard.

*Reviewer #2 (Recommendations for the authors):*

Instead of using the raw data to test a model of dimerization, it is important to build a model of the reporter system to assist in interpreting the data. See the paper describing the development of KTR reporters by Regot et al. (pmid: 24949979) for an excellent example of how this is done and why it is so useful.

There are several other concerns regarding the technology, especially its apparent lack of sensitivity. Maximal biological response to activated EGFR is typically achieved with only a couple thousand occupied receptors per cell (pmid: 29268862; pmid: 27405981). Is this detectable? The results of the HEK293 experiments suggest that this is the case. However, instead of editing those cells, it would be informative to try MCF10A cells, which have hundreds of thousands of endogenous receptors. This could be a very powerful system, especially considering all the live cell imaging studies that have used those cells for understanding EGFR signal transduction.

The use of FRAP is advised to establish the number of pools for the fluorescent reporters and their exchange rates.

Technical on videos:

1. Titles don't agree with supplementary text

2. Text in videos obscured by QuickTime title bar

*Reviewer #3 (Recommendations for the authors):*

I have no specific recommendations for these authors. I felt the paper was clearly written, the experiments were well described, and the methods were extremely detailed.

I feel this paper is well-suited for publication and should be accepted in its present form.

---

## [Author Response]

Overall, the reviewers found your new approach, using RTK-ITAM fusions and fluorescent Syk or Zap70 SH2 domains to monitor RTK activation, to be exciting and innovative. The approach is explained well, and the experiments are clear and clean. However, as you will see from the original reviews and comments below, there was a divergence in opinion on the interpretation of the data and thus on the likely adoption of this new tool by the research community. During the review consultation, several themes emerged:1. Use of different cell types makes it difficult to understand the effect of expression level (of RTK fusion and of SH2 reporter) on the kinetics of probe depletion from the cytosol. There are indications that the assay is capable of being very sensitive, but the use of multiple cell types, expression levels and measurement times makes it very difficult to determine the caveats of the system and the types of applications for which this is most useful. We recommend CRISPR/Cas9 insertion of the ITAM tag into the EGFR gene in the same cell type that is used for over-expression. NIH3T3 cells would perhaps make most sense, since they endogenously express ~40K EGFRs, or MCF10A, where many EGFR kinetics and imaging studies have been done.

We have addressed this theme in two ways.

First, we have added new quantification experiments and simulations to characterize how RTK and ZtSH2 expression levels alter biosensor *activity* (new section “Quantifying reporter activity as a function of pYtag component expression levels” on Lines 191-238; new Figure 1—figure supplements 2-4). We performed these experiments in two cell lines (NIH3T3 and HEK293T) and assessed membrane translocation at a variety of different ZtSH2 and EGFR expression levels. We specifically highlight cases where ZtSH2 levels are varied but EGFR levels are held constant, and *vice versa*. We also modeled these responses using our simple model of ZtSH2 translocation to active EGFR. In brief, we find that biosensor responses depend on the expression levels of both components: ZtSH2 cytosolic clearance is pronounced when RTK expression is high and ZtSH2 expression is low (because these conditions favor complete ZtSH2 translocation to the membrane); conversely, low RTK and high ZtSH2 expression makes it difficult to observe translocation because of a high residual cytoplasmic ZtSH2 pool. These results can be captured nicely in the ratio of RTK:ZtSH2 expression, which does a reasonable job of predicting overall responses (Figure 1—figure supplement 2D and Figure 1—figure supplement 3D). These results also provide some guidance for new users of the system. For example, when studying RTKs expressed at low levels, it is important to establish cells with correspondingly low ZtSH2 expression to observe biosensor responses.

Second, we have now analyzed biosensor *kinetics* as a function of EGFR/ZtSH2 expression levels (Figure 1—figure supplements 2E-F; 3E-F). Here, we computationally ordered single-cell dynamic responses by increasing EGFR/ZtSH2 ratio and plotted the dynamics of ZtSH2 recruitment to the membrane. We broadly find that the kinetics of ZtSH2 membrane translocation are initiated similarly across a broad range of component expression levels, demonstrating that the kinetics of probe depletion are not especially sensitive to the expression levels of our biosensor. Importantly, the same cell line was used for all of our EGF, EREG and EPGN dose-response experiments, so there are also no systematic differences in biosensor expression in these assays where kinetic differences are observed. Finally, we find that EGFR and ErbB2 dynamics are remarkably consistent between cell lines and biosensors when data from all experiments are plotted on top of each other (Figure 5—figure supplement 1), further demonstrating that these differences do not arise because of differences the biosensor construct used (e.g., VISH2 vs ZtSH2) or subtle expression level effects in just one cell line.

We wholeheartedly agree that same-cell-line comparisons between CRISPR-tagged (or lowly expressed) and overexpressed contexts are useful. We chose to compare various EGFR expression levels in HEK293T cells, as suggested by Reviewer 1, due to the challenges and time delays inherent in establishing and validating additional entirely new CRISPR-tagged clonal lines. These data are included in a new Figure 1—figure supplement 3, for comparison to CRISPR results in Figure 6. We find that expression level is indeed an important variable in cellular responses, in line with prior studies showing that EGFR endocytosis is saturatable at high expression levels. CRISPR-tagged HEK293Ts (and low-expressing exogenous variants in Figure 1—figure supplement 3A) exhibit more rapid and complete EGFR endocytosis after stimulation compared to their EGFR-overexpressed counterparts, which maintain high levels of membrane-associated EGFR throughout the timecourse. These data are discussed in Lines 526-536.

In summary, we have addressed the Editor’s concerns by (1) providing new experimental evidence and computational results showing that sensitive biosensor responses can be achieved by scaling ZtSH2 expression to EGFR expression (thereby maintaining a similar EGFR:ZtSH2 ratio); (2) showing that the kinetics of biosensor activation are relatively robust to biosensor expression level; and (3) performing comparisons in the same parental cell line between endogenous receptor levels (via CRISPR/Cas9) and overexpression, which reveal more rapid and pronounced receptor clearance from the plasma membrane at low EGFR expression levels.

2. Discrepancy between apparent sensitivity of the approach (ability to detect EGFR activation using the knock-in approach in HEK 293 cells, in which EGFR expression is thought to be very low), with the inability to detect ERBB2 activation by EGF unless EGFR is co-over-expressed.

This is a very thoughtful point and we are grateful to the reviewers and Editor for bringing it up. Hopefully the results can be better understood now that we have more systematically characterized responses and their variation with component expression level.

For detecting endogenous dynamics of an RTK expressed at low levels, it is imperative to sort for a correspondingly low level of ZtSH2 expression (see e.g. Figure 1—figure supplement 2B or Figure 1—figure supplement 3B). Otherwise, a large residual pool of SH2 domains will lead to an imperceptible change in biosensor concentration! For the CRISPR-tagged data in Figure 6, we paid special attention to this parameter, switching to a bright mScarlet fluorophore and selecting for dimly expressing cells to achieve low but detectable ZtSH2 protein levels (now discussed on Lines 512-524). In contrast, our typical experimental pipeline of Figures 1-5, focusing on overexpressed receptors, was successful because co-transducing with both an RTK and the biosensor led to comparable expression levels of both components. However, these SH2 expression levels were likely too high to detect the small number of active, endogenous EGFR:ErbB2 complexes without EGFR co-overexpression. We hope that the additional data and discussion helps to clarify this point to the reader.

Use of supra-physiological concentrations of ligand (EGF) is also a concern. The sensitivity of the system should be tested with lower EGF concentrations.

We appreciate the reviewers’ and editor’s concern, and completely agree that defining responses at lower EGF doses is important! We carried out a thorough study of the dose-dependency of the system in Figure 3, investigating pYtag responses across nearly 500 cells, 3 ligands and 14 total doses. We include these experiments precisely to demonstrate that pYtags are not solely able to report at supra-physiological doses.

However, we also would like to defend our use of supra-physiological inputs for some experiments in the manuscript. When building and characterizing a new biosensor, it is sensible to begin by examining its response to a saturating input before extending to more challenging cases (low ligand doses; endogenous levels of receptor expression), and we do extend to these challenging use cases in our study. Saturating inputs are also very important to fully interrogate our negative controls, for example to prove that absolutely no ZtSH2 translocation occurs, even at very high ligand doses, when an ITAM-less EGFR is overexpressed. This is a strategy taken in every biosensor development study we are aware of, including Regot *et al*’s ErkKTR paper referred to by one of the reviewers as exemplary, which uses 50 ng/mL anisomycin and 100 ng/mL FGF, or 10 μM forskolin as the sole stimuli for characterizing p38, Erk and PKA responses.

3. Lack of calibration of the experiments. The readout of the system (% clearance) is highly dependent on the ratio of tagged receptor to fluorescent probe, the rate of access of the probe, and the number of internal pools and their rate of exchange. RTK activation will also depend on the ratio of the ligand to receptor as well as ligand affinity. However, it is felt that these parameters have not been properly taken into account or controlled.

There are many questions here, which we will address one by one:

– Ratio of tagged receptor to fluorescent probe: we agree! We hope that this is now comprehensively addressed as described in our response to Theme 1 above (Figure 1—figure supplements 2-4, Lines 191-238).

– Rate of access to the probe (and internal pools): These are very interesting questions, but we are unsure what experiments the editor and reviewers might have in mind to further address them. We have extensive experience over a decade with recruiting proteins from cytosol to membrane (indeed, this is the basis of most tools in cellular optogenetics) and a timescale of 10-20 sec for membrane recruitment is very typical across many contexts, from mammalian cell lines to *Drosophila* embryos (e.g., Toettcher JE et al., *Cell* 2013; Johnson HE et al., *Dev Cell* 2017). We observe membrane translocation on the same timescale here (e.g., Figure 1C, Figure 2A, Figure 3 for EREG and EPGN, which both exhibit a rapid rise to steady-state, and Figure 6D). We also observe minutes-timescale de-recruitment of the fluorescent probe after EGFR inhibition by gefitinib (Figure 1D), indicating that kinetics of both probe recruitment and de-recruitment are fast. In no cases have we visually observed residual probe fluorescence in sub-compartments of the cytoplasm, suggesting that the cytosol acts as a single-compartment reservoir for our probe.

– Ratio of ligand to receptor molecules: This is indeed an important parameter. For our dose-response experiment, we added 150 μL of media containing various ligand concentrations. Assuming a high EGFR expression level of 100,000 receptors per cell, our seeding density of 20,000 cells per well, and a dose of 1 ng/mL EGF, we obtain estimates of 2x10^9^ total EGFR/well and 1.5x10^10^ total EGF ligands/well. Thus, even at low EGF doses, we operate in a regime of excess ligands/receptors, which is appropriate for dose-response measurements. We now discuss these considerations in Lines 931-935.

– Ligand affinity: We consider ligand affinity in our choice of doses used for the dose-response experiments of Figure 3A. We also use the known ligand-binding affinities for EGFR, EREG, and EPGN in our mathematical model. Finally, in the model, we also systematically vary binding affinity using the dimensionless parameter β to assess how it would be predicted to alter response dynamics.

4. The mathematical model makes use of seemingly arbitrary calibration factors rather than using biophysical quantities that have been previously established for the EGFR system. One reviewer questioned whether the mathematical model for dimerization is indeed needed and whether it enhances the message. If the authors feel that the mathematical model should be included, then the authors should consider cleanly calibrating an experiment to better parameterize the math model. If the math model could be used to extract fundamental numbers for the EGFR system, then the same principles could be applied to other ligand-receptor systems analyzed by the pYtag approach.

We would argue strongly in favor of the model’s inclusion because it was instrumental in interpreting the stark differences in EGFR dynamics that we observe in response to different ligands! It also helped us design a validation experiment, where we alter EGFR-EGFR dimerization affinity (Figure 3D) and shift the dynamics of EREG to a more EGF-like response.

However, we agree that the model could have been better explained, particularly in how parameters were obtained from literature or set to match our own experimental data. The base model (for EGF-EGFR binding) contains 9 parameters, 6 of which were obtained from literature (see Tables S1-2 for a list of all parameters and their literature sources).

The remaining three parameters, which reflect forward and reverse rates for receptor dimerization and cross-phosphorylation, were set to match the EGF response kinetics observed in our experiments. The β and γ parameters are simply convenient dimensionless scaling factors whose values reflect the *experimentally determined differences* in binding affinity for EREG/EPGN, which differ in receptor binding by 10-100-fold compared to EGF and produce receptor homodimers with a ~100-fold difference compared to EGF-bound receptors (Freed et al., *Cell* 2017). We thus used values β=50 and γ=100 to represent low-affinity ligands. Similarly, we used experimentally determined values for β and γ to model the 6-fold and 650-fold differences between wild-type EGFR and the GBM mutants for these quantities (Hu et al., *Nature* 2022). Again, we emphasize that all values chosen for our scaling parameters reflect quantitative, experimentally determined differences between high- and low-affinity ligands. We have clarified these points throughout the modeling section of the text (Lines 338-403).

5. Since endocytosis of the tagged EGFR seems slower than reported for endogenous EGFRs in this system, controls are needed to show that tagging does not perturb EGFR signaling or internalization. Some type of EGF uptake or degradation assay in the cells with endogenous EGFR plus/min the tagging system would be an appropriate control.

We thank the reviewers and editors for this insightful comment and have now tested whether endocytosis is different for our pYtagged EGFR compared to a control construct. We have added a new supplementary figure (Figure 2—figure supplement 1) quantifying EGFR membrane intensity and colocalization of EGFR with EEA1, an early endosomal marker, over time after EGF stimulation. We performed this analysis for cells expressing our EGFR-ITAM system versus an ITAM-less EGFR construct expressed at the same level. Broadly, we find that the levels of membrane EGFR and EGFR-EEA1 colocalization are quantitatively indistinguishable (Figure 2—figure supplement 1B-C).

We also now provide data that loss of membrane-localized EGFR is dependent on EGFR expression level, providing some explanation for why endocytosis appears to be slower in our EGFR-overexpressed cells. We imaged HEK293T cells with CRISPR-tagged EGFR, lowly expressed EGFR, and highly expressed EGFR (Figure 1—figure supplement 3 and Figure 6), finding rapid receptor clearance in CRISPR-tagged cells and low overexpression, but not highly overexpressed cells. These data suggest that EGFR expression level, not the presence or absence of the biosensor system, is likely to be the source of differences in endocytosis, consistent with prior reports that EGFR endocytosis is saturatable (e.g. Lund et al., *JBC* 1990). As our goal in this manuscript is to present a biosensor and not study endocytosis, we are glad to report these differences and point out that endogenous expression levels are likely important for obtaining normal degrees of endocytic trafficking for future studies. We describe these results in Lines 254-267 and 526-536.

Overall, it is felt that, as written, the manuscript provides a proof-of-concept rather than making a strong case for others to adapt this assay to study their receptors of interest.

We hope that our more thorough analysis and extension to several additional RTKs will change the perspective of the Reviewers and Editor on this point.

Reviewer #1 (Recommendations for the authors):1. On page 9, the authors state: "Notably, we observed that pYtag-expressing cells stimulated for at least 30 min with EGF contained internalized vesicles that were positive for both total EGFR and ZtSH2 (Figures 2C and 2D). Subsequent treatment with Gefitinib eliminated ZtSH2 from EGFR-positive vesicles within minutes, suggesting that the enrichment of ZtSH2 at vesicles is indicative of signaling from endosomal compartments (Figures 2C and 2D)." To conclude that the accumulation of EGFR and ZtSH2 in punctae that are vesicles/endosomal compartments simultaneous labeling with appropriate markers should be performed (see also below, point 2). The same applies to experiments performed on CRISPR-Cas9 genome-edited HEK 293T cells [page 18: "We also observed rapid and near-complete internalization of endogenous EGFR from the cell membrane, with some internalized vesicles retaining residual ZtSH2 labeling (Figure 6D, right-most panels)."].

We appreciate the reviewer’s comments, which we have addressed in two ways. First, we have added new data demonstrating that EGFR and ZtSH2 indeed colocalize with the early endosome marker EEA1 after EGF stimulation in both NIH3T3 and HEK293T cells (Figure 2—figure supplement 1D-G), as well as for an ITAM-less EGFR control (Figure 2—figure supplement 1A-C).

Second, we altered the wording describing our live-imaging results in Figure 2C-D, Figure 6D, and Video 5, and instead refer to puncta of EGFR and ZtSH2 that move away from the cell surface, likely reflecting internalization (Lines 269-279, 526-536).

2. It would be interesting (and maybe useful to add some novel insights) to follow the receptor routes after it has reached the cell membrane to observe whether the activation also occurs in (or from) specific endocytic compartments (e.g. Rab5 or EEA-1 positive early endosomes, Rab7 or LAMP1 positive late endosomes/lysosomes, Rab11 positive perinuclear/recycling compartment). This set of experiments could be performed by either simultaneously expressing fluorescent protein-tagged endosomal/vesicular markers or staining the corresponding endogenous proteins in cells that were fixed at different time points upon ligand stimulation.

We agree that detailed studies of receptor activity during trafficking is an excellent use case for pYtags, which we hope will be taken up by the receptor trafficking community. To spur this line of thinking, we have now included colocalization for one endosomal marker (EEA1) in Figure 2—figure supplement 1.

3. The experiments over-expressing the pYtags were mainly performed in NIH3T3 cells, whereas the CRISPR-Cas9 knock-in was done in HEK 293T. It would be useful to check whether the same differences would be also seen in HEK 293T over-expressing the tag. After assessing this, further characterization of the signal emitted by the cell line generated by CRISPR-Cas9 would add some key concepts to RTK activity dynamics (e.g. the localization of the RTK activation signal in the genetically manipulated cells in Figure 6, looks quite different from the signal shown in Figure 1. In Figure 6, it looks more dot-like and less localized in cell-to-cell contacts. Do the authors have an explanation for this?).

We thank the reviewer for this excellent suggestion. We have now added data comparing HEK293T cells with the endogenous CRISPR tag, low overexpression, and high overexpression (a new Figure 1—figure supplement 3). Indeed, we see that high EGFR expression correlates with more sustained EGFR membrane localization after EGF stimulation, in line with prior results that the rate of EGFR endocytosis is affected by expression level (Lund et al. *JBC* 1990). We agree with the reviewer’s observation that membrane activation is more punctate in the CRISPR tagged cells! We are not aware of a clear explanation for this interesting phenomenon.

4. Some experiments shown in Figures 1 and 2 were repeated only two times. The analysis of at least three independent experiments is in general a well-accepted standard.

We thank the reviewer for pointing this out. We have carefully checked our notes for Figure 2, and all experiments were performed in biological triplicate (although in one case, two biological replicates were imaged on the same day). We have updated the figure legend accordingly. We have performed an additional repeat of the SYF experiment in Figure 1I-J, which confirms Src-independent ZtSH2 translocation.

While we agree that three replicates are an important standard for virtually all experiments, we would argue that different standards apply to screen designed to generate a lead candidate for further study. For example, a drug screen or directed evolution campaign is often carried out just once, but of course the major hits are characterized in detail in many subsequent triplicate experiments. Our experiment of Figure 1D is such a screen – we tested multiple ITAM sequences with the goal of identifying one for all subsequent experiments. (Indeed, the fact that we observed similar responses in 6 independent cell lines harboring distinct ITAM sequences constitutes very strong proof that ITAMs are capable of potent translocation responses; the question of “do ITAMs function as biosensors” was thus tested in 12 replicates: 6 cell lines x 2 microscopy sessions.) Our resulting lead candidate, the CD3ε ITAM, was then validated extensively in all subsequent experiments in the paper. We argue that this experiment should be held to the same standard as an initial candidate screen, but we are happy to defer to the Editor on this point.

Reviewer #2 (Recommendations for the authors):Instead of using the raw data to test a model of dimerization, it is important to build a model of the reporter system to assist in interpreting the data. See the paper describing the development of KTR reporters by Regot et al. (pmid: 24949979) for an excellent example of how this is done and why it is so useful.

We thank the reviewer for the suggestion. We agree that modeling the cytosol-to-membrane translocation of heterodimerization systems is a good idea and is something we have included in prior optogenetics papers that depend on similar biochemical events (e.g., Toettcher et al., *Nature Methods* 2011). Our existing mathematical model does include the heterodimerization system, making it a good choice for studying the relationship between ZtSH2 and EGFR expression level. We have further added a new supplementary figure (Figure 1—figure supplement 4) in which we systematically examine ZtSH2 translocation in response to different expression levels of EGFR and ZtSH2, which qualitatively agrees with our experimental results (Figure 1—figure supplements 2-3). Broadly, we find that the EGFR-to-ZtSH2 expression ratio is a crucial parameter, which we now explain and which helps inform our CRISPR-tagging strategy in Figure 6.

There are several other concerns regarding the technology, especially its apparent lack of sensitivity. Maximal biological response to activated EGFR is typically achieved with only a couple thousand occupied receptors per cell (pmid: 29268862; pmid: 27405981). Is this detectable? The results of the HEK293 experiments suggest that this is the case. However, instead of editing those cells, it would be informative to try MCF10A cells, which have hundreds of thousands of endogenous receptors. This could be a very powerful system, especially considering all the live cell imaging studies that have used those cells for understanding EGFR signal transduction.

We strongly disagree with the reviewer’s comment that our biosensor lacks sensitivity – as the reviewer points out, the biosensor can detect endogenous EGFR responses in HEK293T cells.

However, we agree that we did an insufficient job in presenting how sensitivity is controlled by experimentally tunable parameters in our system. To clarify these parameters, we now include modeling and data of performance at different expression levels (Figure 1—figure supplements 2-4). Briefly, we find that the EGFR/ZtSH2 expression *ratio* is an essential parameter: for low EGFR expression, correspondingly low ZtSH2 expression is required to avoid leaving a large cytosolic pool of ZtSH2 under conditions of maximum EGFR activation. Conversely, ZtSH2 translocation will be prematurely saturated if it is expressed at a much lower level than that of active EGFR.

The use of FRAP is advised to establish the number of pools for the fluorescent reporters and their exchange rates.

We appreciate the suggestion but unfortunately our imaging setup is poorly suited for FRAP experiments. However, we observe rapid dynamics of cytosol-to-membrane translocation in response to step-up inputs (e.g., low-affinity ligand stimulation; Figure 3A) and step-down experiments (e.g., gefitinib treatment; Figure 1C). These results argue a rapid and dynamic exchange between cytosol and membrane, in concordance with the typical ~20 sec cytosol-to-membrane translocation timescales that we typically observe for optogenetic tools.

Technical on videos:1. Titles don't agree with supplementary text2. Text in videos obscured by QuickTime title bar

We have gone through legends carefully and believe these issues are now fixed.

Reviewer #3 (Recommendations for the authors):I have no specific recommendations for these authors. I felt the paper was clearly written, the experiments were well described, and the methods were extremely detailed.I feel this paper is well-suited for publication and should be accepted in its present form.

We thank the reviewer for their time and enthusiasm!